# Gli1 marks a sentinel muscle stem cell population for muscle regeneration

Jiayin Peng [1,10], Lili Han[2,10], Biao Liu[1,10], Jiawen Song[1], Yuang Wang[1], Kunpeng Wang[1,3], Qian Guo[2], XinYan Liu[1], Yu Li [1], Jujin Zhang[2], Wenqing Wu[1], Sheng Li[4], Xin Fu[4], Cheng-le Zhuang[5], Weikang Zhang[6,7], Shengbao Suo[6,8], Ping Hu [4,5,6,9] ✉ & Yun Zhao [1,2,3] ✉

Adult skeletal muscle regeneration is mainly driven by muscle stem cells (MuSCs), which are highly heterogeneous. Although recent studies have started to characterize the heterogeneity of MuSCs, whether a subset of cells with distinct exists within MuSCs remains unanswered. Here, we find that a population of MuSCs, marked by Gli1 expression, is required for muscle regeneration. The Gli1+ MuSC population displays advantages in proliferation and differentiation both in vitro and in vivo. Depletion of this population leads to delayed muscle regeneration, while transplanted Gli1+ MuSCs support muscle regeneration more effectively than Gli1− MuSCs. Further analysis reveals that even in the uninjured muscle, Gli1+ MuSCs have elevated mTOR signaling activity, increased cell size and mitochondrial numbers compared to Gli1− MuSCs, indicating Gli1+ MuSCs are displaying the features of primed MuSCs. Moreover, Gli1+ MuSCs greatly contribute to the formation of $G_{Alert}$ cells after muscle injury. Collectively, our findings demonstrate that Gli1+ MuSCs represents a distinct MuSC population which is more active in the homeostatic muscle and enters the cell cycle shortly after injury. This population functions as the tissue-resident sentinel that rapidly responds to injury and initiates muscle regeneration.

Stem cells are crucial for homeostasis and post-injury regeneration. They are generally considered residing in niche in the deep quiescent state under physiological condition. Usually, both quiescent and active stem cell subpopulations may coexist in the healthy tissues with rapid cell turnover, such as skin, intestinal epithelium, and hematopoietic tissues, but not in the tissues with slow cell turnover, such as brain and muscle[1,2]. However, recent studies have found a low percentage of activated stem cells in brain, indicting the fact that quiescent and active stem cells may coexist in the tissues and organs with slow cell turnover rate as well[3,4]. It is probably that the low proportion of the

[1]State Key Laboratory of Cell Biology, Shanghai Institute of Biochemistry and Cell Biology, Center for Excellence in Molecular Cell Science, Chinese Academy of Sciences, University of Chinese Academy of Sciences, Shanghai 200031, PR China. [2]Key Laboratory of Systems Health Science of Zhejiang Province, School of Life Science, Hangzhou Institute for Advanced Study, University of Chinese Academy of Sciences, Hangzhou 310024, PR China. [3]School of Life Science and Technology, ShanghaiTech University, 100 Haike Road, Shanghai 201210, PR China. [4]Xinhua Hospital affiliated to Shanghai Jiao Tong University School of Medicine, Shanghai 200023, PR China. [5]The 10th People's Hospital affiliated to Tongji University, Shanghai 200072, PR China. [6]Guangzhou Laboratory-Guangzhou Medical University, Guangzhou 510005, PR China. [7]College of Life Science and Technology, Huazhong University of Science and Technology, Wuhan 430074, PR China. [8]The First Affiliated Hospital of Guangzhou Medical University, State Key Laboratory of Respiratory Disease, Guangzhou, Guangdong 510005, PR China. [9]Key Laboratory of Biological Targeting Diagnosis, Therapy and Rehabilitation of Guangdong Higher Education Institutes, The Fifth Affiliated Hospital of Guangzhou Medical University, Guangzhou, Guangdong 510005, PR China. [10]These authors contributed equally: Jiayin Peng, Lili Han, Biao Liu. ✉e-mail: hup@sibcb.ac.cn; yunzhao@sibcb.ac.cn

primed stem cells remains undiscovered in muscle. Local injury and environmental stress induce the transition of muscle stem cells (MuSCs) to an intermediate state referred to as $G_{Alert}$, which can enter the cell cycle faster and differentiate more efficiently[5,6]. mTOR signaling activity is high in $G_{Alert}$ MuSCs, and hepatocyte growth factor activator (HGFA) can induce the transition from quiescent MuSCs to $G_{Alert}$ MuSCs[5-7]. $G_{Alert}$ cells are considered to be primed stem cells to meet the requirement for the rapid activation of muscle regeneration for muscle injuries. However, the origin of $G_{Alert}$ cells remains unclear.

MuSCs expressing the transcription factor Pax7 are essential for homeostasis and injury repair of skeletal muscle[8-12]. MuSCs usually stay in quiescence and transit from the quiescent state to the activated state under injury or pathological conditions[11,13]. Accumulating evidences from single-cell RNA-sequencing (scRNA-seq) analysis, lineage tracing, transplantation studies, and other assays have suggested that MuSCs are heterogeneous and comprise various subpopulations with different abilities to support muscle regeneration[6,14-19]. MuSC heterogeneity can be defined based on the differential gene expression profiles and cell morphologies[6,14,20-23]. However, the functions of each subpopulation of MuSCs are not clearly understood yet. A better understanding of the heterogeneity of MuSC subpopulations and their functions will help to have deeper insights on muscle regeneration, and provide therapeutic insight into treatment of muscle-related diseases.

Previous studies have demonstrated the pivotal role of the Hedgehog (Hh) signaling pathway in muscle regeneration, involving both canonical and non-canonical mechanisms[24-26]. Gli family of transcription factor play important roles as Hh signaling effectors. Recent study found that Gli3 is required for maintaining MuSCs quiescence, and MuSCs lacking Gli3 enter the $G_{Alert}$ state in the absence of injury via activation of mTORC1 signaling[24]. Gli2 is transiently upregulated upon MuSCs activation, and aging MuSCs are defective for both primary cilium and Gli2 expression at basal state and after acute injury[25]. Gli1, as a direct target gene, plays a role in positive feedback by enhancing Gli activity. Thereby, we have reason to suspect that Gli1 plays an important role in muscle regeneration. Gli1 has been shown to be expressed in the stromal cell of many organs[27-30]. Gli1+ cells function as a microenvironmental cell type to facilitate the repair of tissue injury[27,31-34]. It remains unclear whether Gli1 labels a designated subsets of MuSCs or progenitor cells. And the characterizations of these different Gli1+ cell states under homeostatic conditions are not clear as well.

Here, we identified a Gli1+ MuSC population by scRNA-seq and multiple lineage tracing. Gli1+ MuSCs display superior proliferation ability and are required for regeneration of muscle injury. Further studies revealed that Gli1+ MuSCs showed many properties of $G_{Alert}$ cells, such as increased cell size and mitochondrial metabolism/activity and elevated level of mTOR signaling in uninjured mice, suggesting the existence of a distinct population of alerting MuSCs under physiological conditions. Moreover, Gli1+ MuSCs contributed to unproportionally big percentage of $G_{Alert}$ cells after muscle injury. This study identified a new population of primed MuSCs which is ready to be activated to serve as the sentinels that rapidly respond to muscle injury and speed the regeneration program.

## Results

### Gli1 marks a population of Pax7+ MuSCs

Gli1 has been shown to be an important transcription factor that labels various lineages of cells[29-31,33,35]. To examine the distribution of Gli1-expressing cells in the skeletal muscle, we performed scRNA-seq using Gli1-CreERT2; R26-tdTomato mice, in which Gli1+ cells were labeled by tdTomato (tdT) after tamoxifen (TAM) induction. A total of 8508 tdT+ (Gli1+) cells were sorted from hindlimb skeletal muscles of 3 adult male mice and pooled for scRNA-seq (Fig. 1a, Supplementary Fig. 1a). Seven distinct cell clusters were identified (Fig. 1b). Consistent with the previous reports[31,36], a large number of tdT+ (Gli1+) cells in the skeletal

muscle were fibro/adipogenic progenitors (FAPs), mesenchymal progenitors (MPs), tenocytes and fibroblasts (Fig. 1b, c, Supplementary Fig. 1b-f). In addition to these cell types, we also found that ~5.2% of tdT+ (Gli1+) cells expressed MuSC markers, such as Pax7, suggesting that Gli1 is expressed in MuSCs (Fig. 1d-f). To further confirm the expression of Gli1 in the MuSC population, CD45−CD31−Sca1−Vcam1+ MuSCs were isolated from wild-type mice by fluorescence-activated cell sorting (FACS), and analyzed by scRNA-seq (Fig. 1g, Supplementary Fig. 1g). To confirm the FACS sorting results, Pax7 expression was analyzed, and the majority of the FACS-sorted cells were Pax7+, confirming the proper isolation of MuSCs. The Pax7+ cell population was further analyzed (Fig. 1h). The clustering analysis revealed a population of Gli1+pax7+ cells (Fig. 1i), indicating the presence of a group of MuSCs expressing Gli1.

Hh signaling has been reported to regulate muscle regeneration[25], and Gli1 is the key transcription factor for the activation of Hh signaling. In Gli1-CreERT2 mice, the insertion of CreERT2 leads to a frameshift of Gli1 gene. To avoid the effects caused by the loss of Gli1 expression in homozygous Gli1-CreERT2, we chose to only use mice containing heterozygous Gli1-CreERT2 allele. We also further ensured that Gli1 expression was not disrupted in the heterozygous mice containing Gli1-CreERT2 by RT-qPCR. The expression of Gli1 was not changed (Supplementary Fig. 2a). Skeletal muscle injury was induced in WT and Gli1-CreERT2 mice by intramuscular injection of cardiotoxin (CTX) in the tibialis anterior (TA) muscle, and the regeneration of the injured muscle was examined on 0, 3, 5, 7 and 14 days post injury (dpi). No differences were detected in Gli1-CreERT2 heterozygous mice, confirming no muscle regeneration defects in these mice (Supplementary Fig. 2b). Therefore, we used mice containing heterozygous Gli1-CreERT2 allele for the following experiments.

To validate the presence of Gli1+ MuSCs identified by scRNA-seq in the muscle tissue, we performed a series of lineage-tracing experiments using various mouse models. Gli1-CreERT2; R26-tdT mice were administered oil or TAM. At 3 days, 14 days, or 30 days after induction, tdT+ (Gli1+) cells in muscle were traced to further verify the scRNA-seq results (Fig. 2a). A tdT (Gli1)-expressing subset of MuSCs dispersed throughout the skeletal muscles was observed starting from day 3 after TAM induction, and the percentage of tdT+ (Gli1+) MuSCs plateaued at day 14 after TAM induction, showing no further increase at day 30 (Fig. 2b, c). In contrast, no tdT+ (Gli1+) cells were detected in oil-treated Gli1-CreERT2; R26-tdT mice at day 30 after TAM induction (Fig. 2b, c), indicating the specific induction of tdT in the muscle tissue. Consistent with the previous study[31], immunofluorescence staining for tdT and PDGFRα in TA muscle sections showed that tdT+ (Gli1+) PDGFRα+ cells were detectable in skeletal muscles at day 14 after TAM induction (Fig. 2d, Supplementary Fig. 3a). In addition, tdT+ (Gli1+) Pax7+ cells (~13.8% of Pax7+ cells) were detected in TA muscle sections by immunofluorescence staining (Fig. 2e, f), as well as on single extensor digitorum longus (EDL) myofibers isolated from Gli1-CreERT2; R26-tdT mice (Fig. 2g, Supplementary Fig. 3b, c). These results further confirmed the existence of the Gli1+ MuSC population in vivo.

To exclude the potential contamination from other Gli1+ cells, dual-recombinase-based lineage tracing was performed as described[37]. We first generated a Pax7-DreERT2 knock-in mouse line, and bred it with a R26-RSR-tdT (R26-Rox-Stop-Rox-tdT) mouse line (Supplementary Fig. 4a, b). The efficient induction of Pax7-DreERT2-mediated recombination by TAM was confirmed (Supplementary Fig. 4c). Immunofluorescence staining and FACS analysis on TA muscle sections and MuSCs showed that over 85% of Pax7+ cells were expressing tdT, indicating a high efficiency and specificity of Pax7-DreERT2 for lineage tracing of Pax7+ cells (Supplementary Fig. 4d-f). Next, Gli1-CreERT2; Pax7-DreERT2; Ai66 (Gli1-CreERT2; Pax7-DreERT2; R26-Rox-Stop-Rox-Loxp-Stop-Loxp-tdT) mice were generated to specifically label Gli1+ Pax7+ cells. In this mouse line, expression of the tdT reporter required both Dre-Rox and Cre-LoxP recombination to exclude the

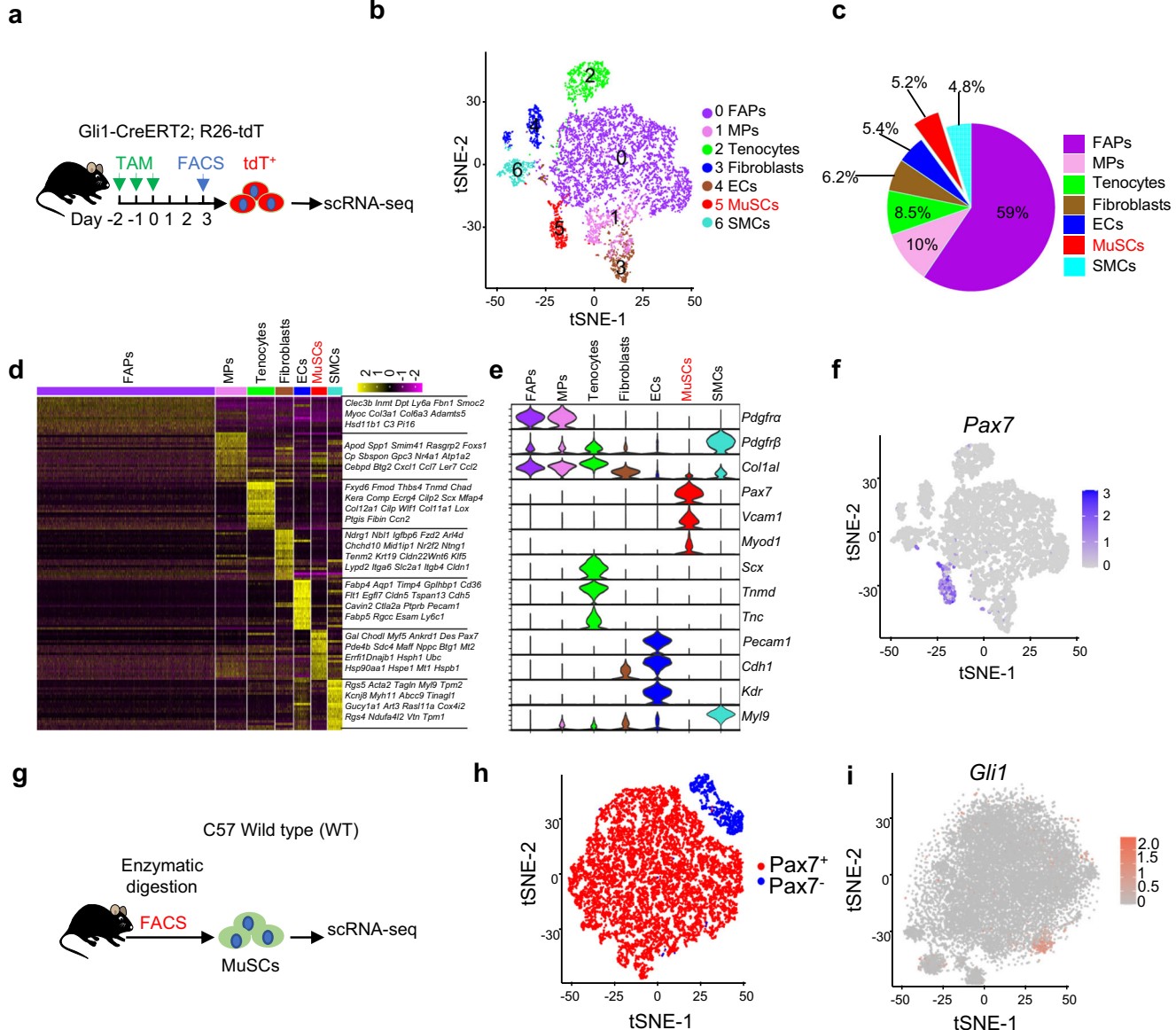

**Fig. 1 | scRNA-seq analysis of Gli1-expressing cells. a** Scheme of the experimental strategy of scRNA-seq. **b** Visualization of unsupervised clustering in a t-distributed stochastic neighbor embedding (t-SNE) plot of 8508 Gli1$^+$ cells isolated from hindlimb skeletal muscles ($n = 3$). **c** Each cell clustering percentage in t-SNE plot of 8,508 cells. Percentage of cells is shown in the pie diagrams. **d** Heatmap of scRNA-seq data showing enriched genes in the 7 different clusters. **e** Violin plots showing the expression levels of representative marker genes across the 7 clusters. **f** Distribution of *Pax7* across all populations. **g** Scheme of the experimental strategy. **h** Visualization of unsupervised clustering in a t-distributed stochastic neighbor embedding (t-SNE) plot of 17,959 MuSCs isolated from hindlimb skeletal muscles ($n = 3$). **i** Distribution of *Gli1* expression across Pax7$^+$ populations.

potential contamination from other Gli1$^+$ cells (Fig. 2h, i). Dual-recombinase tracing revealed that ~7.9% of MuSCs were tdT$^+$ (Gli1$^+$ Pax7$^+$) in skeletal muscles after TAM induction (Fig. 2j, k, Supplementary Fig. 4g, h). In contrast, no tdT$^+$ (Gli1$^+$ Pax7$^+$) cells were detected in the littermate controls (Gli1-CreERT2; Ai66 or Pax7-DreERT2; Ai66) (Fig. 2k, Supplementary Fig. 4g, h). Because dual-recombinase-mediated lineage tracing depends on both inducible CreERT2 and DreERT2, the efficiency would be lower than that from single lineage tracing[37]. Consistently, there were fewer tdT$^+$ (Gli1$^+$ Pax7$^+$) cells labeled in Gli1-CreERT2; Pax7-DreERT2; Ai66 mice than in Gli1-CreERT2; R26-tdT mice (Fig. 2k). These results suggested that a Gli1$^+$ population of MuSCs exists in skeletal muscle.

Another potential confounding factor in this lineage-tracing approach is that the observed tdT$^+$ cells could have been derived from the progeny of tissue-resident stem or progenitor cells that previously expressed Gli1[38,39]. To exclude this possibility, the Gli1−LacZ

mouse line, where nuclear LacZ is knocked in the Gli1 locus, was used to ascertain whether tdT$^+$ cells actively express Gli1. Cryosections from skeletal muscle of Gli1−LacZ mice were analyzed by X-Gal staining and immunofluorescence staining. X-Gal staining further confirmed the presence of Gli1$^+$ cells in skeletal muscles (Supplementary Fig. 5a). Consistent with the previous report[31], Gli1 expression was detected in PDGFRα$^+$ FAPs (Supplementary Fig. 5b). Meanwhile, colocalization of β-Gal and Pax7 was also detected in skeletal muscle (Supplementary Fig. 5c). About 11.6% of Pax7$^+$ MuSCs actively express Gli1 (Supplementary Fig. 5c), further confirming the existence of a population of Gli1$^+$ MuSCs in skeletal muscle (Fig. 2l).

**Gli1$^+$ MuSCs display enhanced proliferation and differentiation ability in vitro**

To further characterize the differences between Gli1$^+$ and Gli1$^−$ MuSCs, these cells were isolated and immunofluorescence stained for tdT and

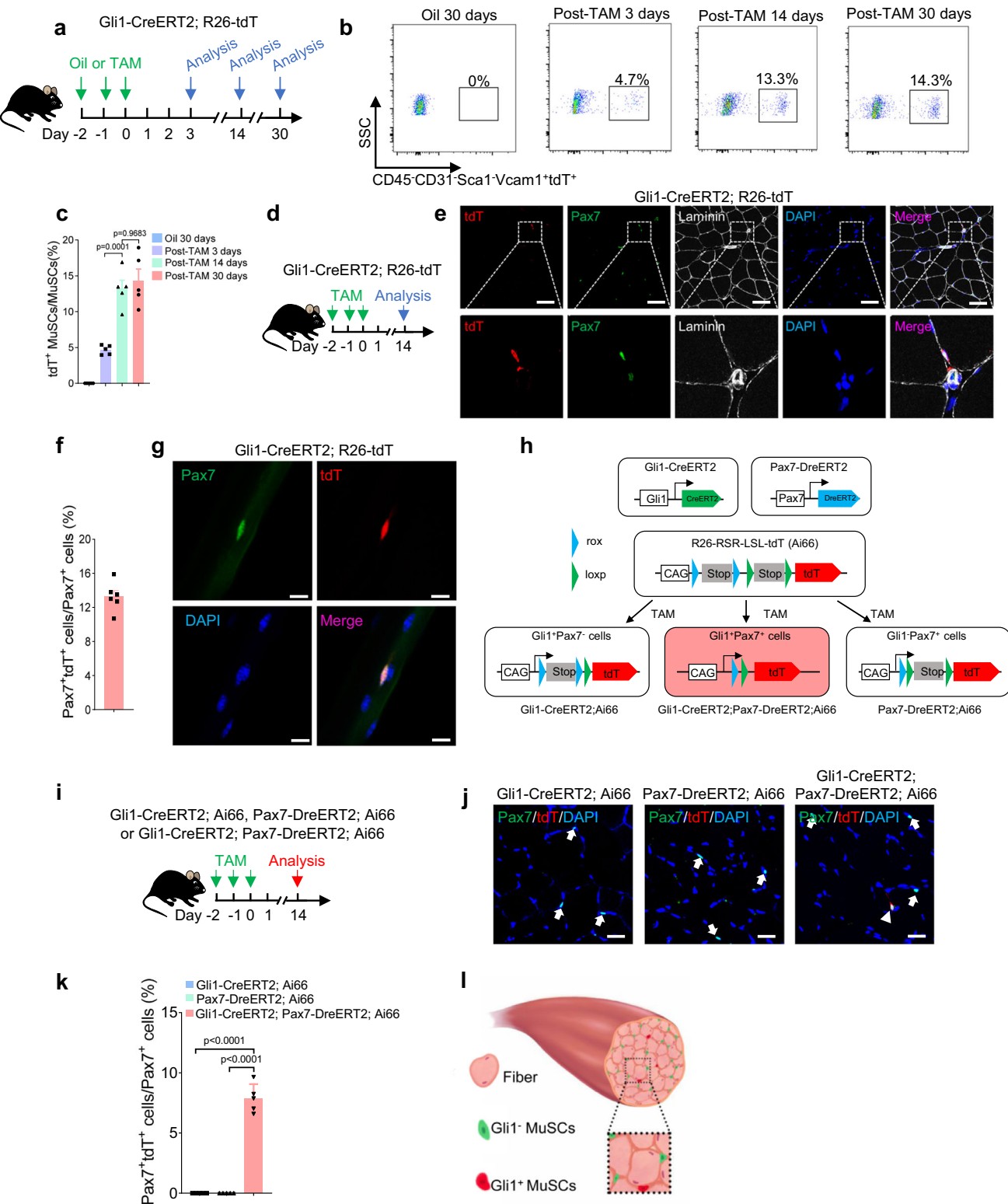

Gli1, Pax7, Myf5, CD34 or MyoD (Fig. 3a). Immunofluorescence staining results showed that more than 95% of tdT⁺ (Gli1⁺) and tdT⁻ (Gli1−) MuSCs expressed Pax7, and no statistically significant difference in Pax7, Myf5, CD34 and MyoD protein expression between tdT⁺ (Gli1⁺) and tdT⁻ (Gli1⁻) MuSCs (Fig. 3b, c, Supplementary Fig. 6a). However, Gli1 is predominantly expressed in tdT⁺ (Gli1⁺) MuSCs (Fig. 3b), which was further identified by Western blot analysis result (Fig. 3d). Subsequently, we separately collected tdT⁺ (Gli1⁺) and tdT⁻ (Gli1−) MuSCs for RNA-Seq analysis (Supplementary Fig. 6b). We identified 208

differentially expressed genes, of which 178 were upregulated and 30 were downregulated in tdT⁺ (Gli1⁺) MuSCs compared to tdT⁻ (Gli1⁻) MuSCs (Fig. 3e). RNA-sequencing analysis reveals that tdT⁺ (Gli1⁺) MuSCs exhibited high expression levels of *Gli1* and *Hhip*, while *Gli3* is barely expressed (Supplementary Fig. 6c). There was no significant difference in the expression level of *Ptch1* (Supplementary Fig. 6c). In addition, tdT⁺ (Gli1⁺) and tdT⁻ (Gli1⁻) MuSCs showed no significant differential expression of *Pax7/MyoD/Myf5/CD34/MyoG* (Supplementary Fig. 6c). We also performed a standard RT-qPCR to validate the

**Fig. 2 | Identification of a population of MuSCs marked by Gli1. a** Scheme of the experimental strategy. **b** Representative profiles of fluorescence-activated cell sorter analysis of the percentage of Gli1$^+$ MuSCs from Gli1-CreERT2; R26-tdT mice at different time points with oil or TAM treatment ($n$ = 5). tdT-marked Gli1$^+$ MuSCs were detected within 3 days and plateaued at day 14, showing no further increase at day 30. **c** Quantification analysis of the percentage of MuSCs expressing tdT at different time points with oil or TAM treatment ($n$ = 5). **d** Scheme of the experimental strategy for analyzing TAM-induced Gli1$^+$ MuSCs. **e** Immunofluorescence staining for tdT (red), Pax7 (green), Laminin (gray) and DAPI (blue) in TA muscles ($n$ = 6). Scale bars, 50 μm. Boxed regions are magnified on the low. **f** Quantification of the percentage of Pax7$^+$ cells expressing tdT (Gli1) ($n$ = 6). **g** Immunofluorescence staining for tdT (red), Pax7 (green) and DAPI (blue) in isolated single EDL myofibers from hindlimb skeletal muscles ($n$ = 5). Scale bars, 10 μm. **h** Scheme of the dual-recombinase genetic lineage-tracing strategy. **i** Schematic of the experimental strategy. **j** Immunofluorescence staining for tdT (red), Pax7 (green) and DAPI (blue) in TA muscle sections. White arrowhead indicated the tdT$^+$ (Gli1$^+$) Pax7$^+$ cells from different mouse lines ($n$ = 5). White arrows indicated the tdT$^-$ (Gli1$^-$) Pax7$^+$ cells. Scale bars, 50 μm. **k** Quantification of the percentage of Pax7$^+$ cells expressing tdT (Gli1) in different mouse lines ($n$ = 5). **l** The model specific of MuSC populations. Red indicated the Gli1$^+$ MuSCs. Green indicated Gli1$^-$ MuSCs. Light red indicated myofibers. Data are presented as mean ± SEM; Statistical significance was determined by one-way ANOVA (**c, k**). All numbers ($n$) are biologically independent experiments. Source data are provided in the Source Data File.

gene expression results observed from sequencing experiment. The expression pattern of these genes was in concordance with the sequencing results (Supplementary Fig. 6d, e). Next, GO analysis revealed that genes involved in proliferation and differentiation were upregulated in tdT$^+$ (Gli1$^+$) MuSCs (Fig. 3f, g), indicating that Gli1$^+$ MuSCs have advanced proliferation and differentiation capabilities.

The proliferation ability of Gli1$^+$ and Gli1$^-$ MuSCs was next tested. The same amount of tdT$^+$ (Gli1$^+$) and tdT$^-$ (Gli1$^-$) MuSCs were seeded separately. After 4 days of in vitro culturing, tdT$^+$ (Gli1$^+$) MuSCs displayed higher proliferative capability as indicated by the elevated cell numbers (Supplementary Fig. 7a, b). Similarly, the results from the 5-Ethynyl-2'-Deoxyuridine (EdU) labeling experiment suggested that Gli1$^+$ MuSCs proliferated more extensively than Gli1$^-$ (tdT$^-$) MuSCs (Fig. 3h). To further investigate the differences in proliferation ability between Gli1$^+$ and Gli1$^-$ MuSCs, CD31$^-$CD45$^-$Sca1$^-$Vcam1$^+$ MuSCs (including tdT$^+$ (Gli1$^+$) and tdT$^-$ (Gli1$^-$) MuSCs) from Gli1-CreERT2; R26-tdT mice were isolated by FACS sorting and cultured in vitro. The cells were passaged once. In this in vitro culturing system, both tdT$^+$ (Gli1$^+$) and tdT$^-$ (Gli1$^-$) MuSCs were cultured in the same mixture to exclude the potential variations in the culturing conditions. The percentage of tdT$^+$ (Gli1$^+$) cells was measured by FACS analysis at each passage (Supplementary Fig. 7c, d). The percentage of tdT$^+$ (Gli1$^+$) cells continued to increase after passage (Supplementary Fig. 7e). To further confirm the proliferation capacity of Gli1$^+$ MuSCs under ex-vivo conditions, muscle fibers from Gli1-CreERT2; R26-tdT mice were isolated and cultured. Samples were collected at 0, 24, 48, and 72 h time points. Immunofluorescence staining of Pax7 revealed that both Gli1$^+$ and Gli1$^-$ MuSCs expressed Pax7 at all time points (Supplementary Fig. 7f). Furthermore, we performed immunofluorescence staining of Gli1 and MyoG on fibers at each time point. The results showed that only tdT$^+$ MuSCs expressed Gli1 and maintained its expression from 0 h to 72 h (Supplementary Fig. 7g). Additionally, we observed only weak MyoG signals in the 72 h samples (Supplementary Fig. 7h). Upon analyzing fibers at 72 h, we observed a significantly higher number of Gli1$^+$ MuSCs compared to Gli1$^-$ MuSCs per clone (Supplementary Fig. 7i). Taken together, these results suggested that Gli1$^+$ MuSCs possess a stronger proliferation capacity.

The differentiation ability of Gli1$^+$ and Gli1$^-$ MuSCs was then examined. Equal numbers of tdT$^+$ (Gli1$^+$) and tdT$^-$ (Gli1$^-$) MuSCs were seeded and induced to differentiate in vitro, respectively. tdT$^+$ (Gli1$^+$) MuSCs displayed a higher fusion index and the proportion of muscle cells with a higher number of cell nuclei is greater than tdT$^-$ (Gli1$^-$) MuSCs (Fig. 3i, Supplementary Fig. 7j). We also have incorporated western blot data of differentiated tdT$^+$ (Gli1$^+$) and tdT$^-$ (Gli1$^-$) for MyHC, MyoD and MyoG. Upon differentiation of MuSCs, we observed low expression levels of MyoD between tdT$^+$ (Gli1$^+$) and tdT$^-$ (Gli1$^-$) myotubes, whereas the protein levels of MyHC and MyoG were significantly higher in tdT$^+$ (Gli1$^+$) myotubes compared to tdT$^-$ (Gli1$^-$) myotubes (Fig. 3j), suggesting the enhanced differentiation ability of Gli1$^+$ MuSCs in vitro.

## Gli1$^+$ MuSCs are critical for proper muscle regeneration in vivo

We next further tested the functions of Gli1$^+$ MuSCs in vivo using Gli1-CreERT2; R26-tdT mice. CTX-induced muscle injury was then generated in the Gli1-CreERT2; R26-tdT mice (Fig. 4a). As indicated by tdT imaging and immunofluorescence staining for laminin in TA muscle sections collected on various days post injury, over 80% of the regenerated myofibers were tdT$^+$ (Gli1$^+$) at 14 dpi (Fig. 4b, c), suggesting that Gli1$^+$ MuSCs actively participate in muscle regeneration. The participation of Gli1$^+$ MuSCs in muscle regeneration was further confirmed by using the dual-recombinase mouse line to rule out the potential contamination from the non-MuSC Gli1$^+$ cells. Injury was induced in TA muscle of Gli1-CreERT2; Pax7-DreERT2; Ai66 mice by intramuscular injection of CTX. Muscle regeneration was monitored by immunofluorescence staining of TA muscle cryosections harvested on 0, 3, 5, 7 and 14 dpi (Fig. 4d). Consistent with the results in Gli1-CreERT2; R26-tdT mice, approximately 70% of the regenerated muscle fibers were from tdT$^+$ (Gli1$^+$) MuSCs (Fig. 4e, f), further confirming that Gli1$^+$ MuSCs contribute to muscle regeneration. To assess the self-renewal capacity of Gli1$^+$ MuSCs, we obtained frozen sections of Gli1-CreERT2; Pax7-DreERT2; Ai66 mice tissue at 14 days post-CTX injury and performed Pax7 staining. The results indicated that the number of Gli1$^+$ MuSCs accounted for approximately 17.4% of the total MuSCs, which was higher than the pre-injury proportion of approximately 13.8%, demonstrating the self-renewal ability of Gli1$^+$ MuSCs (Fig. 4g). These data suggested that Gli1$^+$ MuSCs actively participate the muscle regeneration process.

To further address the function of Gli1$^+$ MuSCs in skeletal muscle regeneration, we generated a Gli1-CreERT2; R26-DTA mouse line to deplete all Gli1$^+$ cells, such as FAPs and fibroblasts together with Gli1$^+$ MuSCs, in vivo by TAM induction (Fig. 5a). After TAM administration, the number of tdT$^+$ (Gli1$^+$) cells decreased in the muscle (Supplementary Fig. 8a–c). Muscle regeneration was surveyed in these mice by inducing injury through CTX injection 7 days after TAM administration, and analyzing TA muscles at 14 dpi (Fig. 5b). The size and weight of the TA muscles were significantly decreased in mice depleted of Gli1$^+$ cells (Fig. 5c). Laminin staining of TA muscle cryosections 14 dpi revealed muscle regeneration defects after depletion of Gli1$^+$ cells (Fig. 5d). These results suggested that Gli1$^+$ cells are required for muscle regeneration.

In Gli1-CreERT2; R26-DTA mice, all Gli1$^+$ cells, such as FAPs and fibroblasts together with Gli1$^+$ MuSCs were eliminated by DTA upon TAM administration. Cell transplantation experiment is a powerful method to specifically examine the effects of Gli1$^+$ MuSCs. However, Gli1$^-$ MuSCs from Gli1-CreERT2; R26-tdT mice cannot be tracked in vivo due to the lack of a fluorescent reporter. To achieve specific discrimination between Gli1$^-$ and Gli1$^+$ MuSCs in vivo, we took advantage of a new genetic lineage-tracing system that utilizes Dre and Cre recombinases to examine Gli1$^-$ and Gli1$^+$ MuSCs. We generated Gli1-CreERT2; R26-eGFP; Pax7-DreERT2; R26-RSR-tdT mice. As shown in the design, Dre expression driven by the Pax7 promoter resulted in tdT gene expression, and Gli1-derived Cre led to GFP gene expression (Supplementary Fig. 8d). Therefore, Gli1$^+$ MuSCs were labeled with GFP

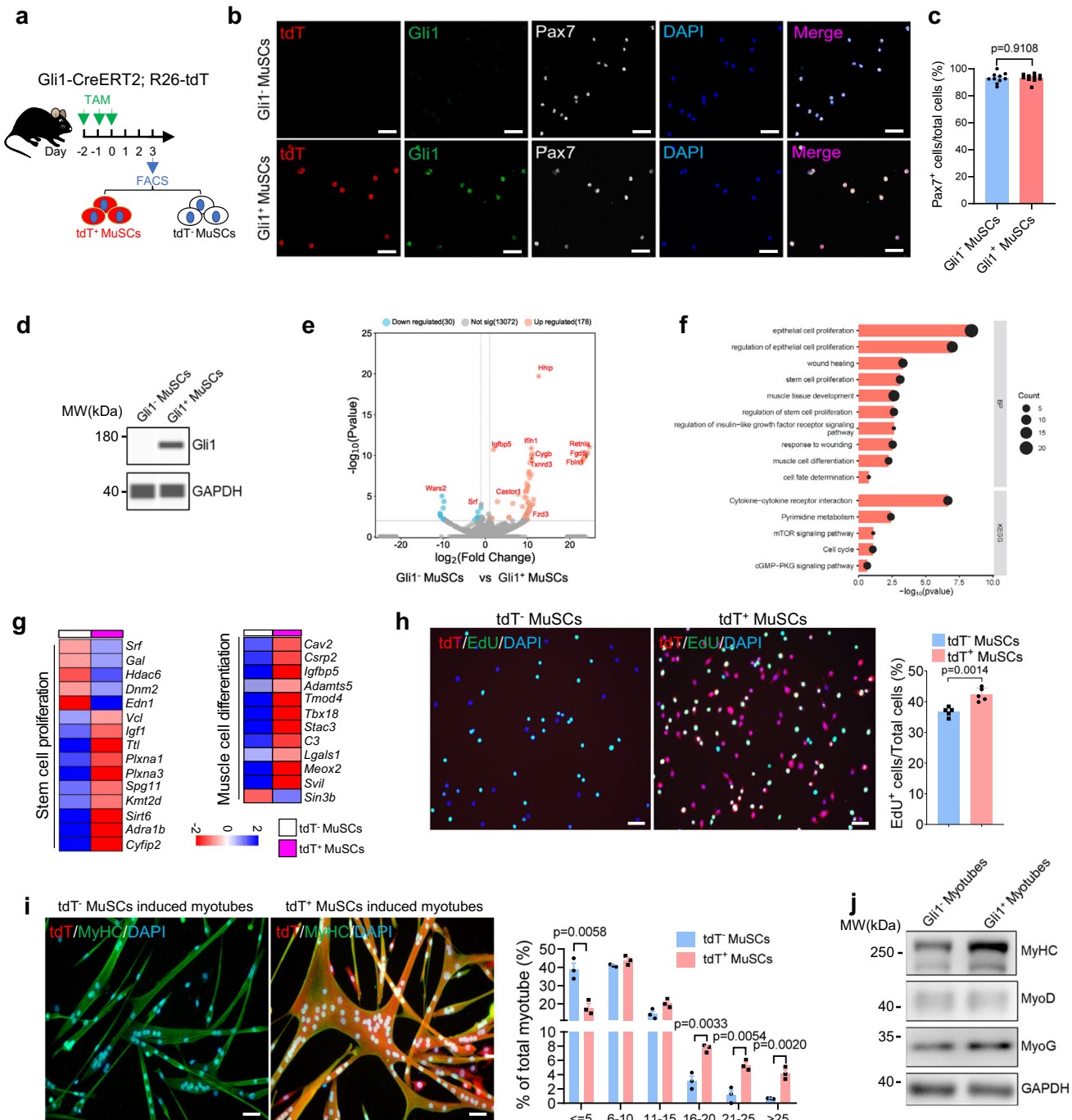

**Fig. 3 | Gli1⁺ MuSCs display enhanced proliferation and differentiation ability in vitro. a** Scheme of the experimental strategy. **b** Immunofluorescence staining for tdT (red), Gli1 (green), Pax7 (gray) and DAPI (blue) on MuSCs isolated from hindlimb skeletal muscles (*n* = 10). Scale bars, 20 μm. **c** Quantification of the percentage of tdT⁺Pax7⁺ cells (*n* = 10). **d** Western blot for Gli1 from tdT⁺(Gli1⁺) and tdT⁻(Gli1⁻) MuSCs. **e** Volcano diagram of differentially expressed genes (red: upregulated; blue: downregulated) in tdT⁻ (Gli1⁻) (*n* = 3) *vs* tdT⁺ (Gli1⁺) (*n* = 4) MuSCs. Not significantly changed genes were indicated in gray. Red and blue highlighted log₂ fold changes of 1 and −1. *p* value < 0.01. **f** Top enriched pathway of the unique genes in tdT⁺ (Gli1⁺) MuSCs. **g** Heatmap of the representative genes in the GO term "cell population proliferation" and "muscle cell differentiation" of tdT⁺ (Gli1⁺) or tdT⁻ (Gli1−) MuSCs. **h** Left panel: Equal numbers (1000 cells) of FACS-sorted tdT⁺ (Gli1⁺) or tdT⁻(Gli1⁻) MuSCs were cultured for 72 h and labeled with EdU

for 12 h, followed by immunofluorescence staining for tdT (red), EdU (green) and DAPI (blue) (*n* = 5). Scale bars, 50 μm. Right panel: Quantification of the percentage of EdU⁺ cells (n = 5). **i** Left panel: Equal numbers (10,000 cells) of tdT⁺ (Gli1⁺) or tdT⁻ (Gli1−) MuSCs were cultured for 2 h in proliferation medium followed by 2 days in differentiation medium; the degree of differentiation was assessed by immunofluorescence staining for tdT (red), MyHC (green) and DAPI (blue) (*n* = 3). Scale bars, 50 μm. Right panel: Quantifications of the percentage of nuclei in MyHC⁺ myotubes (*n* = 3). **j** Western blot for MyHC, MyoD, MyoG and GAPDH (loading control) from tdT⁺(Gli1⁺) and tdT⁻(Gli1⁻) MuSCs induced myotubes. Data are presented as mean ± SEM; Statistical significance was determined by two-tailed unpaired Student's *t* test (**c**, **h**) or one-way ANOVA (**i**). All numbers (*n*) are biologically independent experiments. Source data are provided in the Source Data File.

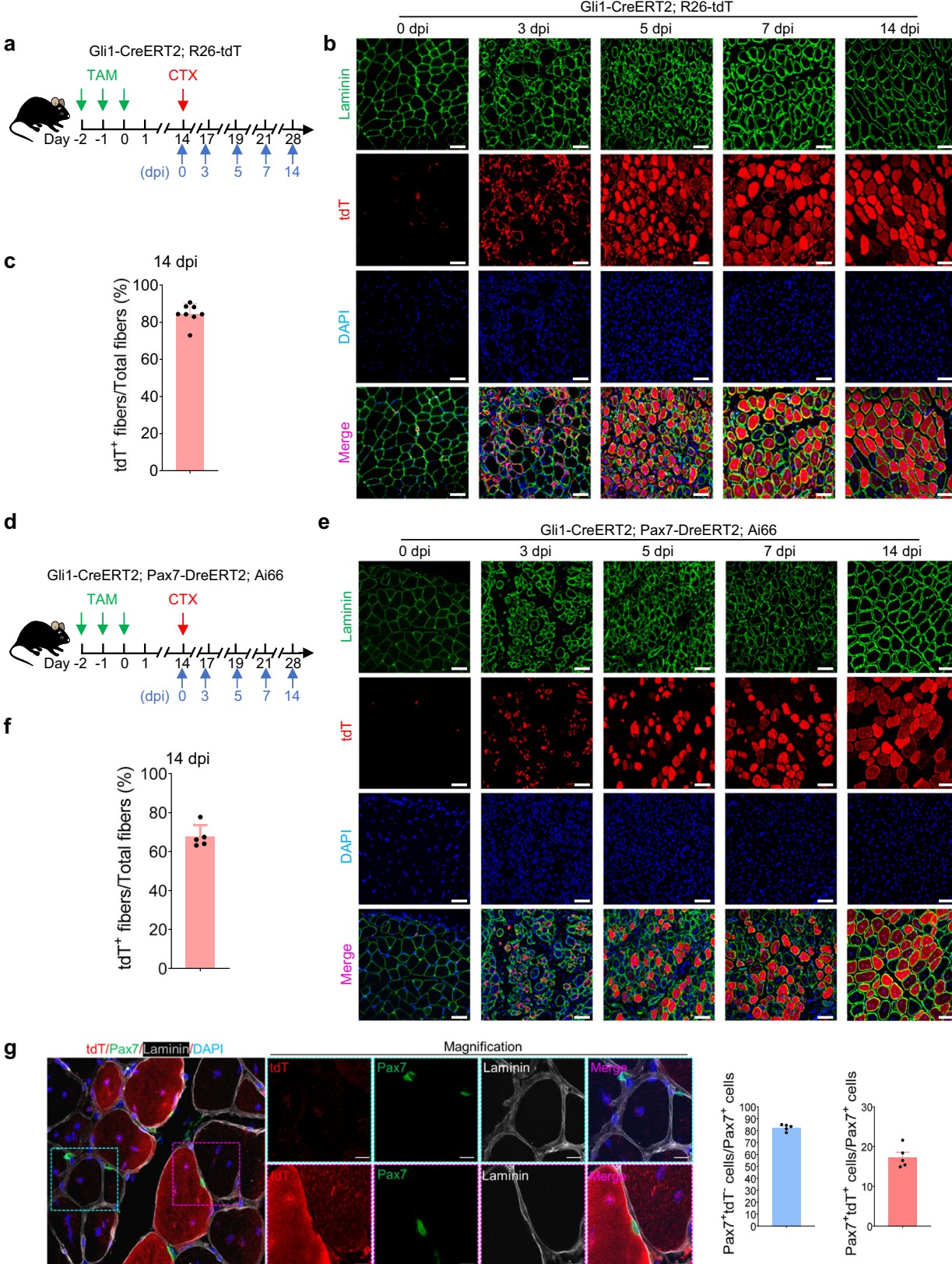

and tdT, while Gli1− MuSCs were only labeled with tdT. Next, we performed genetic lineage-tracing experiments using Gli1-CreERT2; R26-eGFP; Pax7-DreERT2; R26-RSR-tdT mice by administering TAM. We observed the presence of GFP⁺tdT⁺ (Gli1⁺) MuSCs in the muscle, and the proportion of GFP⁺tdT⁺ (Gli1⁺) cells is consistent with percentage in Gli1-CreERT2; R26-tdT mice (Supplementary Fig. 8e, f). To specifically

examine the effects of Gli1⁺ MuSCs, equal numbers (5000 cells) of freshly isolated GFP⁺tdT⁺ (Gli1⁺) and GFP⁻tdT⁺ (Gli1⁻) MuSCs as well as equal volume PBS were transplanted into injured TA muscle of TAM-treated Gli1-CreERT2; R26-DTA mice, in which endogenous Gli1⁺ cells were depleted by DTA (Fig. 5e). Regeneration of TA muscle was evaluated 5, 7 and 14 dpi by laminin staining. Mice transplanted with

**Fig. 4 | Gli1⁺ MuSCs actively participate the muscle regeneration process.**
**a** Scheme of the experimental strategy. **b** Immunofluorescence staining for tdT (red), Laminin (green) and DAPI (blue) in TA muscle sections at different time points after CTX injury in Gli1-CreERT2; R26-tdT mice (*n* = 8). dpi, days post injury. Scale bars, 50 μm. **c** Quantification of the percentage of the regenerated myofibers expressing tdT (*n* = 8). **d** Scheme of the experimental strategy. **e** Immunofluorescence staining for tdT (red), Laminin (green) and DAPI (blue) on TA muscle sections at different time points after CTX injury in Gli1-CreERT2; Pax7-DreERT2; Ai66 mice (*n* = 5). dpi, days post injury. Scale bars, 50 μm. **f** Quantification of the percentage of the regenerated myofibers expressing tdT at 14 dpi (*n* = 5). **g** Left panel: Immunofluorescence staining and enlarged images

for tdT (red), Pax7 (green), Laminin (gray) and DAPI (blue) in TA muscle sections at day 14 after CTX injury in Gli1-CreERT2; Pax7-DreERT2; Ai66 mice (*n* = 5). The boxed regions are magnified in the panels on the right with split channels. The blue boxed region represents Pax7⁺tdT⁻ cells. The pink boxed region represents Pax7⁺tdT⁺ cells. Scale bars, 10 μm. Right panel: Quantification of the percentage of Pax7⁺tdT⁺ and Pax7⁺tdT⁻ cells at 14 dpi (*n* = 5). The red histogram represents the percentage of Pax7⁺tdT⁺ cells in Pax7⁺ cells. The blue histogram represents the percentage of Pax7⁺tdT⁻ cells in Pax7⁺ cells. Data are presented as mean ± SEM. All numbers (*n*) are biologically independent experiments. Source data are provided in the Source Data File.

GFP⁺tdT⁺ (Gli1⁺) MuSCs displayed better regeneration, as indicated by higher cross-sectional area of myofibers than that of mice transplanted with GFP⁻tdT⁺ (Gli1⁻) MuSCs (Fig. 5f, Supplementary Fig. 8g), suggesting that Gli1⁺ MuSCs have greater regenerative ability than Gli1⁻ MuSCs. Given that a substantial generation of newly formed muscle fibers occurs around 5 days post transplantation, we collected TA samples at this time point (5 and 7 dpi) and performed eMHC and Pax7 staining on frozen sections. These results demonstrated a higher number of eMHC⁺tdT⁺ muscle fibers and Pax7⁺tdT⁺ MuSCs in the TA transplanted with Gli1⁺ MuSCs (Fig. 5g, h), providing further evidence of the efficient promotion of skeletal muscle regeneration by Gli1⁺ MuSCs in vivo.

The limitation of using Gli1-DTA mice is the elimination of all Gli1-expressing cells, including MuSCs, FAPs, and other cell types. To specifically investigate the role of Gli1⁺ MuSCs in muscle regeneration while excluding the influence of other Gli1⁺ cells (Gli1⁺ other cells, Gli1⁺ non-MuSCs), we utilized Gli1-CreERT2; R26-tdT mice and Gli1-CreERT2; R26-eGFP; Pax7-DreERT2; R26-RSR-tdT mice as donors. We then sorted tdT⁺ (Gli1⁺) MuSCs from Gli1-CreERT2; R26-tdT mice and GFP⁺tdT⁻ (Gli1⁺) other cells (Gli1⁺ non-MuSCs) from Gli1-CreERT2; R26-eGFP; Pax7-DreERT2; R26-RSR-tdT mice, with a ratio of approximately 1:4 between Gli1⁺ MuSCs and Gli1⁺ other cells. Subsequently, we transplanted these cells into Gli1-CreERT2; R26-DTA recipient mice in a 1:4 ratio of Gli1⁺ MuSCs to Gli1⁺ other cells, resulting in four experimental groups: PBS group (20 μL), Gli1⁺ other cells group (20 μL, 4 ×10⁴ cells), Gli1⁺ MuSCs group (20 μL, 1 × 10⁴ cells), and Gli1⁺ cells group (20 μL, comprising 1 × 10⁴ Gli1⁺ MuSCs and 4 × 10⁴ Gli1⁺ other cells) (Fig. 5i). TA muscles were collected at 14 dpi and cryosections were stained with laminin. The results revealed that TA transplanted with Gli1⁺ MuSCs alone showed robust muscle, with numerous tdT⁺ muscle fibers present, similar to the level of repair observed in the group of Gli1⁺ MuSCs plus Gli1⁺ other cells (Fig. 5j, k). These experimental findings further supported the crucial role of Gli1⁺ MuSCs in the process of muscle regeneration in vivo.

In the above experiments, Gli1⁺ FAPs were depleted that hindered muscle regeneration. To further examine the contribution of Gli1⁺ MuSCs to skeletal muscle regeneration under normal microenvironment in vivo, we performed cell transplantation experiments in immune deficient NOD-SCID mice. The TA muscles of NOD-SCID mice were irradiated to eliminate the resident MuSCs as described previously[40]. Equal numbers of GFP⁺tdT⁺ (Gli1⁺) and GFP⁻tdT⁺ (Gli1⁻) MuSCs (5,000 cells) as well as equal volume PBS were then transplanted into the TA muscle at 1 day after CTX injection, and the TA muscles were harvested 30 days after transplantation (Fig. 6a). We observed that, compared to PBS and GFP⁻tdT⁺(Gli1⁻) MuSCs, GFP⁺tdT⁺ (Gli1⁺) MuSCs extensively participated in the regeneration of the injured TA muscles (Fig. 6b). More pronounced size and weight increases were observed in the muscles of mice transplanted with GFP⁺tdT⁺ (Gli1⁺) MuSCs than in those of mice transplanted with GFP⁻tdT⁺ (Gli1⁻) MuSCs (Fig. 6b), suggesting that Gli1⁺ MuSCs provide greater support for muscle regeneration. To assess muscle function after transplantation, the recipient mice were subjected to downhill treadmill exercises. Running time and distance to exhaustion were significantly longer in mice transplanted with GFP⁺tdT⁺(Gli1⁺) MuSCs

than in mice that received GFP⁻tdT⁺ (Gli1−) MuSCs or PBS transplantation (Fig. 6c). Laminin staining of TA muscle cryosections at 30 dpi in NOD-SCID mice showed that around 482 fibers were contributed by GFP⁺tdT⁺ (Gli1⁺) MuSCs during the regeneration process, whereas GFP⁻tdT⁺ (Gli1⁻) MuSCs contributed to approximately 224 fibers. Consistent with our previous findings, these results demonstrate the enhanced regenerative capacity of Gli1⁺ MuSCs in muscle repair (Fig. 6d, e). Next, Pax7 staining was performed with muscle sections harvested 30 days after transplantation. Immunofluorescence staining for Pax7 showed that nuclear-localized Pax7 expression was observed in the tissue sections of the two groups of recipient mice (Supplementary Fig. 8h). However, we found that there was no significant difference in the number of Pax7-expressing cells, indicating that the two cell populations may have similar self-renewal ability. Although Gli1⁺ MuSCs can contribute more myofibers during regeneration, which indicates stronger proliferation and differentiation abilities, there is no difference in self-renewal ability between Gli1⁺ and Gli1⁻ MuSCs. Taken together, these results provided further evidences that Gli1⁺ MuSCs have greater in vivo muscle regenerative capacity than Gli1⁻ MuSCs.

We next used Duchenne muscular dystrophy (DMD) mice, which lacks the expression of Dystrophin gene, as transplant recipients. 5000 GFP⁺tdT⁺ (Gli1⁺) or GFP⁻tdT⁺ (Gli1⁻) MuSCs⁺ were then transplanted into the TA muscle after CTX injection, respectively (Fig. 6f). The TA muscles were harvested 30 days after transplantation, and the expression of Laminin and Dystrophin were examined by immunofluorescence staining. Laminin staining of TA muscle cryosections at 30 dpi in DMD mice showed that around 341 fibers were contributed by GFP⁺tdT⁺ (Gli1⁺) MuSCs during the regeneration process, whereas GFP⁺tdT⁺ (Gli1⁻) MuSCs contributed to approximately 108 fibers (Fig. 6g, h). Both GFP⁺tdT⁺ (Gli1⁺) and GFP⁻dT⁺ (Gli1⁻) MuSCs formed myofibers expressing Dystrophin (Fig. 6i, j). However, more Dystrophin-expressing myofibers were present in the mice that received Gli1⁺ MuSCs (Fig. 6i, j). Taken together, these results suggested that Gli1⁺ MuSCs have higher ability to support muscle regeneration.

## Gli1⁺ MuSCs are in an alert state

In Gli1⁺ MuSCs, genes related to mTOR signaling were upregulated (Figs. 3f, 7a and Supplementary Fig. 9a). The activation of mTOR signaling in Gli1⁺ MuSCs was further confirmed by measuring the level of phosphorylated S6 ribosomal protein (p-S6), a readout of mTOR signaling[41]. The level of p-S6 was higher in tdT⁺ (Gli1⁺) MuSCs compared to that in tdT⁻ (Gli1⁻) MuSCs (Fig. 7b, c). To further determine the activation of the mTOR signaling pathway in Gli1⁺ MuSCs, we utilized Gli1-CreERT2; R26-tdT mice and isolated their muscle fibers. Immediately after separation, the muscle fibers were fixed with 4% PFA for immunofluorescence staining of p-S6 protein. The results revealed that only Gli1⁺ MuSCs expressed p-S6 (Supplementary Fig. 9b). Additionally, global protein synthesis rate was measured based on puromycin incorporation and examined using Western blotting. The results indicated that compared to Gli1⁻ MuSCs, Gli1⁺ MuSCs exhibited higher protein synthesis levels (Supplementary Fig. 9c), further supporting the notion that Gli1⁺ MuSCs display elevated mTOR signaling. It

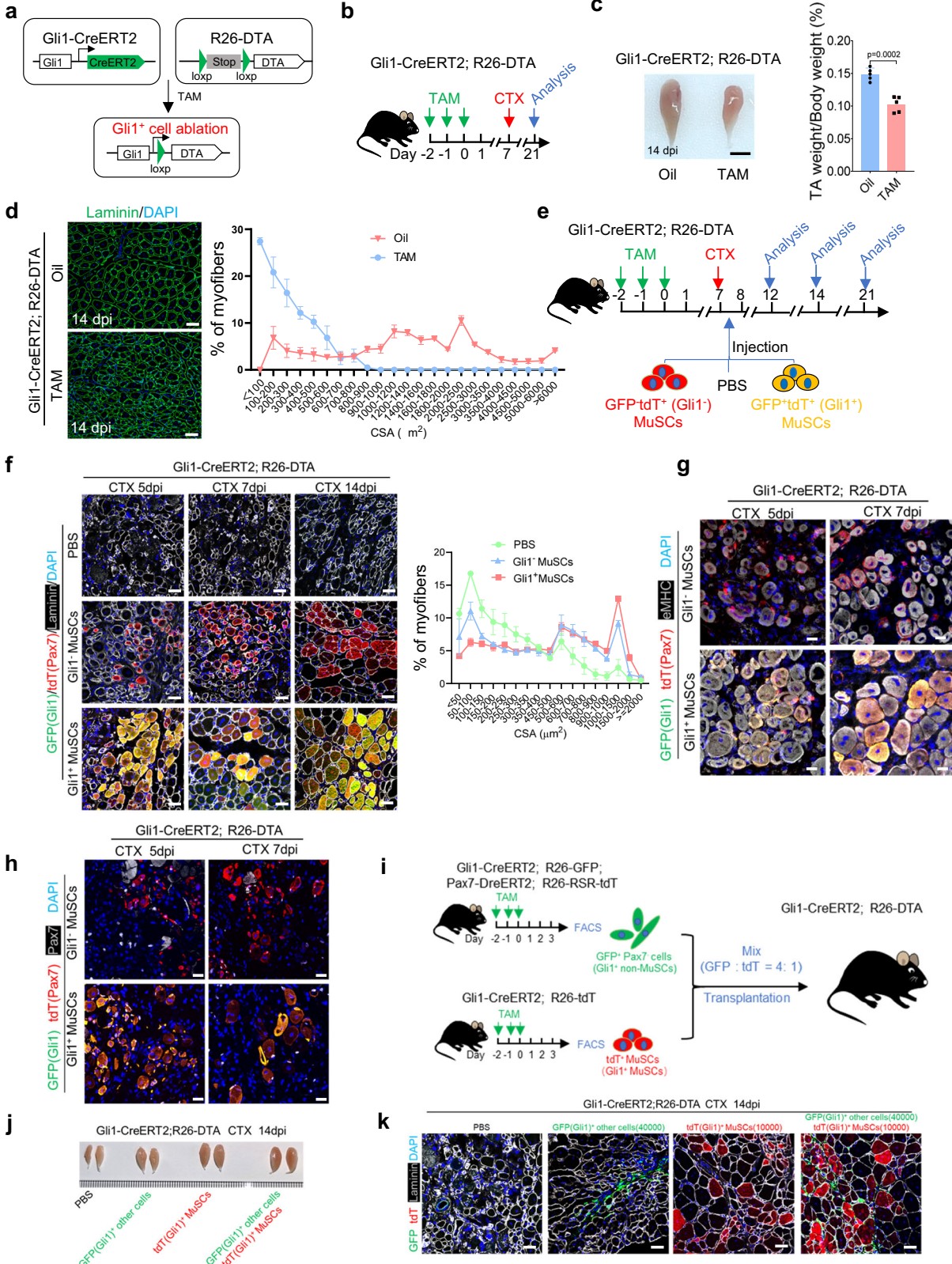

has been reported previously that the up-regulation of mTOR signaling is necessary and sufficient for the alert response of MuSCs to speed damage responses. The "alerting" MuSCs stay in $G_{Alert}$ status display elevated mTOR signaling level[5–7]. We further examined whether Gli1[+] MuSCs have the characteristics of $G_{Alert}$ cells. The cell size of tdT[+] (Gli1[+]) MuSCs was larger than that of tdT[−] (Gli1[−]) MuSCs even in the

uninjured muscle (Fig. 7d). In addition, tdT[+] (Gli1[+]) MuSCs had greater mitochondrial mass than tdT[−] (Gli1[−]) MuSCs (Fig. 7e). Together, Gli1[+] MuSCs display the similar cell morphology as $G_{Alert}$ cells. We next investigated whether Gli1[+] MuSCs enter the cell cycle earlier than Gli1[−] MuSCs after activation in vitro. MuSCs were isolated from Gli1-CreERT2; R26-tdT mice and cultured as described previously[40]. The

**Fig. 5 | Gli1⁺ MuSCs are key contributors to muscle regeneration. a** Scheme of genetic ablation of Gli1⁺ cells. **b** Scheme of the experimental strategy. **c** Left panel: Representative photographs of the TA muscle from Gli1-CreERT2; R26-DTA mice treated with oil or TAM (*n* = 5). Scale bars, 3 mm. Right panel: Ratio analysis of TA muscle weight to body weight (*n* = 5). **d** Left panel: Immunofluorescence staining for Laminin (green) and DAPI (blue) in TA muscle sections from Gli1-CreERT2; R26-DTA mice treated with oil or TAM (*n* = 3). Scale bars, 50 μm. Right panel: quantification of the myofiber cross-sectional area (CSA, μm²) of the CTX-injured TA muscles of Gli1-CreERT2; R26-DTA mice treated with oil or TAM (*n* = 3). **e** Scheme of the experimental strategy. **f** Left panel: Representative images of the GFP⁻tdT⁺ (Red, Gli1−) and GFP⁺tdT⁺ (Yellow, Gli1⁺) fibers from TA muscle 5 days, 7 days, and 14 days after transplantation (*n* = 3). Scale bars, 50 μm. Right panel: Quantification of the myofiber cross-sectional area (CSA, μm²) (*n* = 3). **g** Immunofluorescence staining for GFP (green), tdT (red), eMHC (gray) and DAPI (blue) in CTX-injured Gli1-CreERT2; R26-DTA mice were shown (*n* = 3). Scale bars, 50 μm. **h** Immunofluorescence staining for GFP (green), tdT (red), Pax7 (gray) and DAPI (blue) in CTX-injured Gli1-CreERT2; R26-DTA mice were shown (*n* = 3). Scale bars, 50 μm. **i** Scheme of the experimental strategy. **j** Representative photographs of the TA muscle from Gli1-CreERT2; R26-DTA mice 14 days after transplantation (*n* = 3). Scale bars, 3 mm. **k** Immunofluorescence staining for tdT (red), GFP (green), Laminin (gray) and DAPI (blue) in CTX-injured Gli1-CreERT2; R26-DTA mice were shown (*n* = 3). Scale bars, 50 μm. Data are presented as mean ± SEM; Statistical significance was determined by two-tailed unpaired Student's *t* test (**c**) or one-way ANOVA (**f**). All numbers (*n*) are biologically independent experiments. Source data are provided in the Source Data File.

time for tdT⁺ (Gli1⁺) and tdT⁻ (Gli1⁻) MuSCs to enter the first cell cycle was assayed by live-cell imaging. The Gli1⁺ MuSCs underwent the first cell division after seeding earlier than Gli1⁻ MuSCs (Fig. 7f). In agreement with this finding, EdU was incorporated into more tdT⁺ (Gli1⁺) MuSCs than tdT⁻ (Gli1⁻) MuSCs within the first 40 h after isolation from TA muscles (Fig. 7g). Hence, these findings demonstrated that Gli1⁺ MuSCs mimic G$_{Alert}$ cells and are prone to be activated even in uninjured muscles.

We then further characterized the dynamic changes in Gli1⁺ MuSCs during muscle regeneration. The numbers of Gli1⁺ and Gli1− MuSCs during muscle regeneration were surveyed using Gli1-CreERT2; R26-tdT or Gli1-CreERT2; Pax7-DreERT2; Ai66 mice, in which Gli1⁺ MuSCs were labeled by tdT; while Gli1− MuSCs were non-fluorescent. The percentage of tdT⁺(Gli1⁺) MuSCs was quantified by FACS from muscles at 0, 3, 5, 7 and 14 dpi. We found that the percentage of tdT⁺(Gli1⁺) MuSCs peaked at day 3 after CTX-induced muscle injury and returned approximately to the baseline level after 7 dpi (Fig. 7h, Supplementary Fig. 9d-f, Supplementary Table 1), suggesting that Gli1⁺ MuSCs undergo rapid expansion in the early stage of muscle regeneration in vivo and may represent the cell population to be activated earlier.

The above results suggested that Gli1 marks a unique MuSC population (Figs. 1 and 2). To further distinguish whether the Gli1⁺ MuSCs are a distinct cell population or an intermediate cell status, we further examined whether the more dormant Gli1⁻ MuSCs can convert to Gli1⁺ MuSCs. tdT⁺ (Gli1⁺) and tdT⁻ (Gli1⁻) primary MuSCs isolated from Gli1-CreERT2; R26-tdT mice were seeded in plates. 4-Hydroxytamoxifen (4-OHT) or the same amount of ethanol which is the solvent of 4-OHT was added to induce the Cre driven by Gli1. If Gli1⁺ MuSCs were not a distinct cell population and were an intermediate cell status, the Gli1⁻ MuSCs will gradually turn to tdT⁺ cells. After 4 passages, the Gli1− MuSCs remain to be tdT⁻ (Fig. 7i). Next, we performed transplantation experiments on NOD-SCID mice. TAM was injected intraperitoneally on the second day after tdT⁻ (Gli1⁻) MuSCs transplanting (Supplementary Fig. 10a). After TAM treatment, we did not observe tdT+ fibers or cells in the receipient mice, indicating that tdT⁻ (Gli1⁻) MuSCs do not convert to tdT⁺ (Gli1⁺) MuSCs in vivo (Supplementary Fig. 10b). These results suggested that Gli1⁺ MuSCs represent a distinct population which is not converted from Gli1⁻ MuSCs.

The above results suggested that Gli1⁺ MuSCs are ready to be activated even in uninjured muscle, mimicking the G$_{Alert}$ MuSCs which are reported to present in the muscle located at the counter side of the local muscle injury[5–7]. Next, we investigated whether Gli1⁺ MuSCs contribute to G$_{Alert}$ MuSCs at the counter side muscle after injury. Muscle injury was induced by CTX at the right TA in Gli1-CreERT2; R26-tdT mice. The percentage of tdT⁺ (Gli1⁺) MuSCs in the TA located at the contralateral leg (left TA) of the injured leg (contralateral MuSCs, CSC) was measured by tdT FACS analysis at day 1 after injury (Fig. 8a, b). tdT⁺ (Gli1⁺) MuSCs display rapid cell-cycle entry, increased proliferation and elevated mTOR signaling in the contralateral TA muscle

(Supplementary Fig. 10c-h). In addition, the CSC percentage was compared with the percentage of MuSCs in TA muscle from uninjured mice. As expected, and consistent with the previous reports, the number of MuSCs increased significantly in the contralateral TA muscle of the injured mice compared to that in the uninjured mice (Fig. 8c, Supplementary Fig. 10i). Gli1⁺ MuSCs accounted for over 24% of the MuSCs in the contralateral side of the injured leg. In sharp contrast, only about 13.6% of MuSCs were Gli1⁺ in the uninjured mice (Supplementary Fig. 10j). Strikingly, the over 50% of the newly generated MuSCs induced by the injury at the contralateral leg was Gli1⁺ MuSCs (Fig. 8d). These results suggested that Gli1⁺ MuSCs are sensitive to activation cues and function as rapid responding cells after injury to make major contribution to muscle regeneration.

## Discussion

Considerable heterogeneity exists within the MuSC population. The identification and characterization of MuSC subset with specific functions would bring important insights into the mechanism that regulates muscle regeneration. Here, we identified a Gli1⁺ MuSC population by scRNA-seq and genetic lineage tracing, and discovered that Gli1⁺ MuSCs contribute directly to muscle injury repair. Selective Gli1⁺ cell ablation in muscle coupled with MuSC transplantation showed that Gli1⁺ MuSCs have superior regenerative ability than Gli1⁻ MuSCs. Further investigation showed that Gli1⁺ MuSCs stay in an "alert" state in uninjured muscle, and display rapid response to activation cues. These primed cells are a distinct population of MuSCs which serves as sentinel cells to enable rapid response after injury and speed muscle regeneration (Fig. 8e).

Substantial evidences have shown that both quiescent and active stem cell subpopulations coexist in tissues with rapid cell turnover, such as hematopoietic system, intestine, and hair follicles[1,2,42]. In these tissues, the coexistence of quiescent and active stem cells enables them to respond more swiftly to injury and other external stimuli to accelerate damage repair and maintain the stem cell pool at the same time[43,44]. Nevertheless, it has been generally considered that all stem cells remain quiescence in tissues with slow cell turnover rate. However, with the development of new techniques, many minor cell subpopulations have been identified and the small number of less quiescent stem cells in the tissues with slow turnover rate start being discovered. For example, a recent study reported the presence of stem cells in the state of shallow quiescence in the brain, indicating that deep quiescent and shallow quiescent (prone to be activated) stem cells may also coexist in the tissues with slow cell turnover rate[3,4]. Indeed, two muscle stem cell stages in the homeostatic skeletal muscle have been reported. The deep quiescent "genuine" muscle stem cell marked by CD34$^{High}$ and the more active "primed" MuSCs converted from CD34$^{Low}$ MuSCs by FoxO inactivation have been identified in uninjured muscle[23]. These findings revealed that the conversion from the deep quiescent status to the primed status occurs in homeostatic skeletal muscle. Here, we found another population of "prone-to-active" MuSCs in the homeostatic skeletal muscle marked by Gli1. In

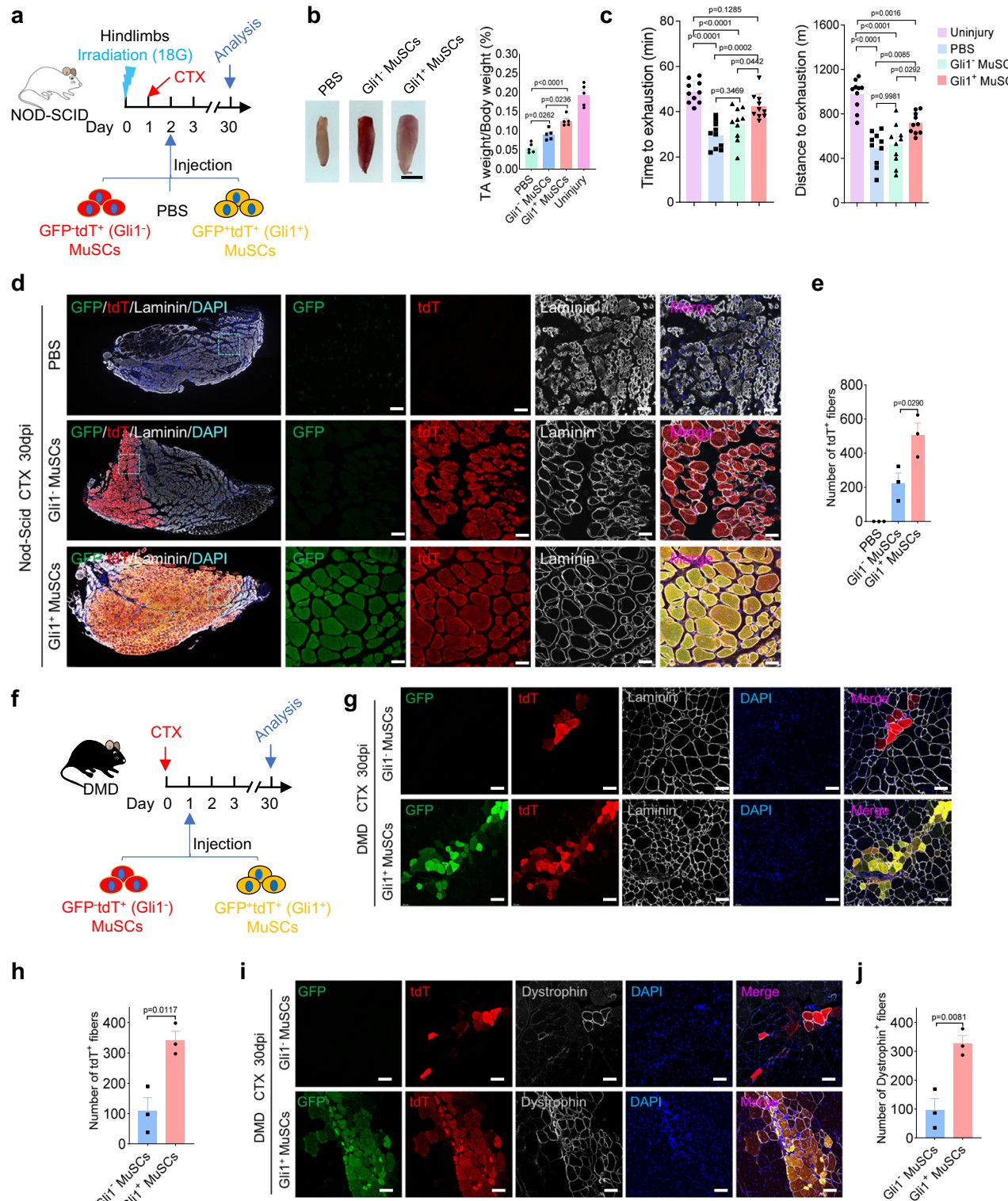

contrast to the primed CD34[Low] MuSCs, Gli1[+] MuSCs are not converted from the more quiescent Gli1[−] MuSCs. They stably present in the homeostatic skeletal muscle to serve as sentinels and contribute to the first wave of cells being activated by injury.

$G_{Alert}$, first reported in MuSCs stimulated by injury at the contralateral leg, is an intermediate cell-cycle stage between $G_0$ and $G_1$[5]. The transition from a deep quiescent state ($G_0$) to a "primed" quiescent state ($G_{Alert}$) during injury and stress enables stem cells to proliferate and differentiate more rapidly upon stimulation[6]. However, it remains unclear whether there is a subset of MuSCs that stay in the $G_{Alert}$ state

under homeostatic conditions. Recently, it was reported that depletion of Gli3 in MuSCs increased the expression of *Gli1*, *Ptch1* and other Gli target genes, induced mTOR signaling activation, and facilitated the entry of MuSCs into the $G_{Alert}$ state[24]. Though the majority of MuSCs in uninjured mice are not expressing Gli1, we identified a small population of MuSCs expressing Gli1 naturally present in uninjured muscle. These cells showed the features of $G_{Alert}$, consistent with the observations in Gli3 KO MuSCs. In our experiments, we observed that Gli1 is predominantly expressed in freshly isolated Gli1[+] MuSCs. The expression of Gli3 was barely detected in Gli1[+] MuSCs, while its

**Fig. 6 | Gli1⁺ MuSCs have higher ability to support muscle regeneration.**
**a** Scheme of the experimental strategy. Equal numbers (5000 cells) of freshly iso-
lated GFP⁺tdT⁺ (Gli1⁺) MuSCs and GFPtdT⁺ (Gli1⁻) MuSCs from Gli1-CreERT2; R26-
eGFP; Pax7-DreERT2; R26-tdT mice, along with the same volumes of PBS were
transplanted into pre-injured TA muscle of the NOD-SCID mice. **b** Left panel:
Representative photographs of the TA muscles from NOD-SCID mice (*n* = 5). Scale
bars, 2 mm. Right panel: Statistical analysis of TA muscle weight to body weight
(*n* = 5). PBS injection served as control. **c** Time (Left panel) and distance (Right
panel) of exhaustion in a downhill treadmill running exercise were measured in
mice that received transplanted GFP⁺tdT⁺ (Gli1⁺) or GFP⁻tdT⁺ (Gli1⁻) MuSCs (*n* = 10).
**d** Immunofluorescence staining for tdT (red), GFP (green), Laminin (gray) and DAPI
(blue) on NOD-SCID mice TA muscle sections (*n* = 3). The boxed regions are mag-
nified in the panels on the right with split channels. Scale bars, 50 μm.
**e** Quantification analysis of the number of tdT⁺ fiber in the recipient NOD-SCID mice

(*n* = 3). **f** Scheme of the experimental strategy. Equal numbers (5000 cells) of
freshly isolated GFP⁺tdT⁺ (Gli1⁺) MuSCs and GFPtdT⁺ (Gli1⁻) MuSCs from Gli1-
CreERT2; R26-eGFP; Pax7-DreERT2; R26-tdT mice were transplanted into CTX-
injured TA muscle of the DMD mice. **g** Immunofluorescence staining for tdT (red),
GFP (green), Laminin (gray) and DAPI (blue) on TA muscle sections (*n* = 3). Scale
bars, 50 μm. **h** Quantification analysis of the number of tdT⁺ fiber in the recipient
DMD mice (*n* = 3). **i** Immunofluorescence staining for tdT (red), GFP (green), Dys-
trophin (gray) and DAPI (blue) on TA muscle sections (*n* = 3). Scale bars, 50 μm.
**j** Quantification analysis of the number of Dystrophin⁺ fiber in the recipient DMD
mice (*n* = 3). Data are presented as mean ± SEM; Statistical significance was deter-
mined by one-way ANOVA (**b**, **c**, **e**) or two-tailed unpaired Student's *t* test (**h**, **j**). All
numbers (*n*) are biologically independent experiments. Source data are provided in
the Source Data File.

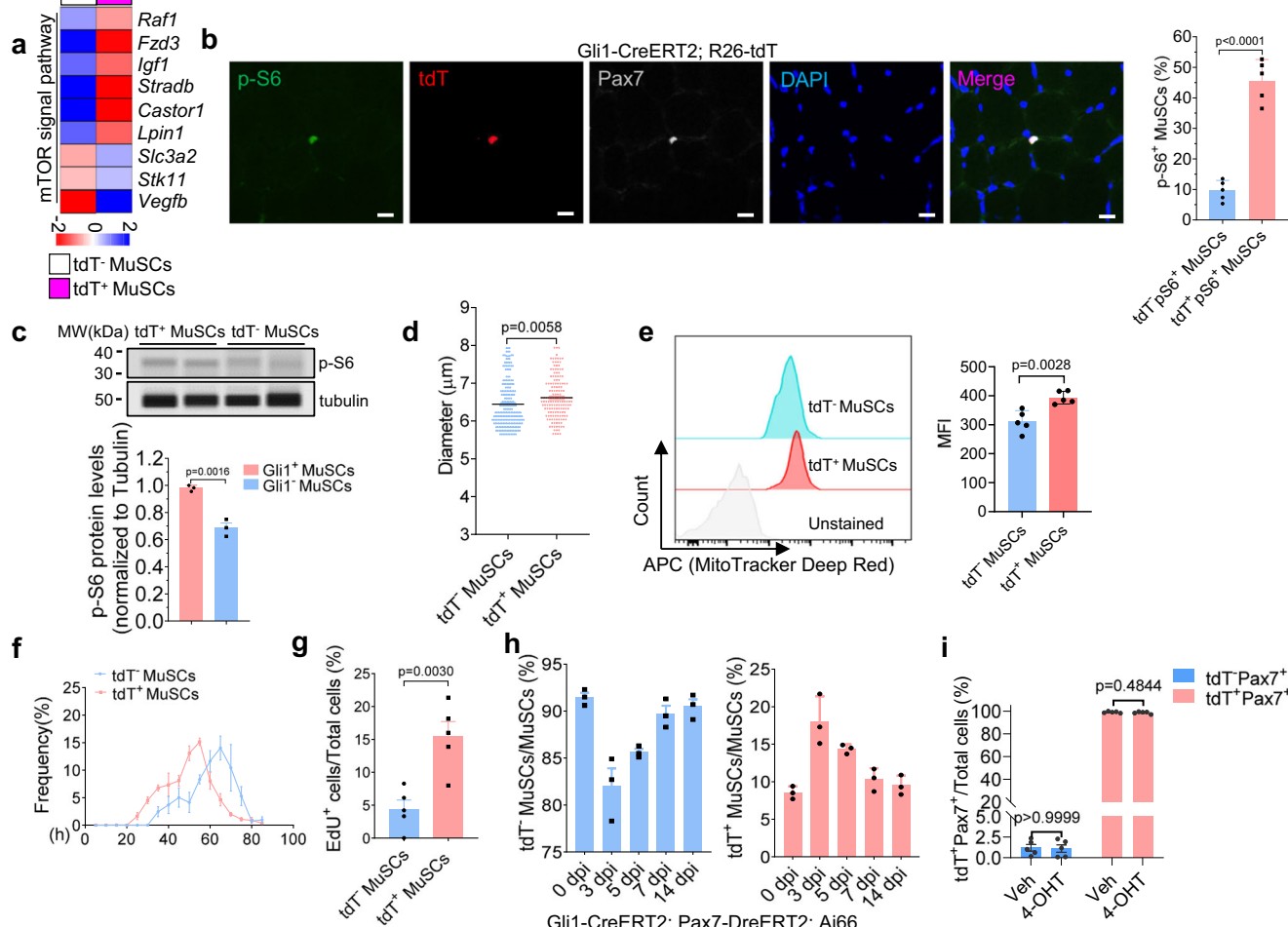

**Fig. 7 | Gli1⁺ MuSCs are in an alert state. a** Heatmap of the representative genes in
the GO term "mTOR signaling pathway" of tdT⁺ (Gli1⁺) or tdT⁻ (Gli1⁻) MuSCs. **b** Left
panel: Immunofluorescence staining of p-S6 (green), tdT (red), Pax7 (gray) and
DAPI (blue) in Gli1-CreERT2; R26-tdT mice TA muscle sections (*n* = 5). Scale bars,
10 μm. Right panel: Quantification of the percentage of tdT⁺ (Gli1⁺) and tdT⁻ (Gli1⁻)
MuSCs expressing p-S6 (*n* = 5). **c** Western blot for p-S6 and Tubulin (loading con-
trol) from tdT⁺(Gli1⁺) and tdT⁻ (Gli1⁻) MuSCs using a capillary-based western blot
automated system. Bar graph indicated the p-S6 protein levels normalized to
Tubulin (*n* = 3). **d** The diameters of MuSCs were measured using Coulter Multisizer
4e (*n* = 1000 cells from 3 mice per group). **e** Left panel: Representative plots of tdT⁺
(Gli1⁺) and tdT⁻ (Gli1⁻) MuSCs stained with MitoTracker Deep Red (MTDR) (*n* = 5).
Right panel: Quantifications of the MFI (Mean fluorescence intensity) of tdT⁺ (Gli1⁺)
and tdT⁻ (Gli1⁻) MuSCs (*n* = 5). **f** Time to the first division in tdT⁺ (Gli1⁺) and tdT⁻

(Gli1⁻) MuSCs was measured by time-lapse microscopy (*n* = 60 cells from 3 mice per
group). **g** Quantification of the percentage of tdT⁺ (Gli1⁺) and tdT⁻ (Gli1⁻) MuSCs
with incorporated EdU. Equal numbers of FACS-sorted tdT⁺ (Gli1⁺) and tdT⁻ (Gli1⁻)
MuSCs were cultured and labeled with EdU for 40 h (*n* = 5). **h** Quantification of the
percentage of tdT⁺ (Gli1⁺) and tdT⁻ (Gli1⁻) MuSCs from Gli1-CreERT2; Pax7-DreERT2;
Ai66 mice at different time points after CTX injury (*n* = 3). dpi, days post injury.
**i** Equal numbers of FACS-sorted tdT⁺ (Gli1⁺) or tdT⁻ (Gli1⁻) MuSCs were cultured with
0.1 μM 4-Hydroxytamoxifen (4-OHT) or the same amount of ethanol for 4 passages.
The proportion of tdT⁺ (Gli1⁺) MuSCs was counted (n = 5). Data are presented as
mean ± SEM; Statistical significance was determined by two-tailed unpaired Stu-
dent's *t* test (**b**, **c**, **d**, **e**, **g**, **i**). All numbers (*n*) are biologically independent experi-
ments. Source data are provided in the Source Data File.

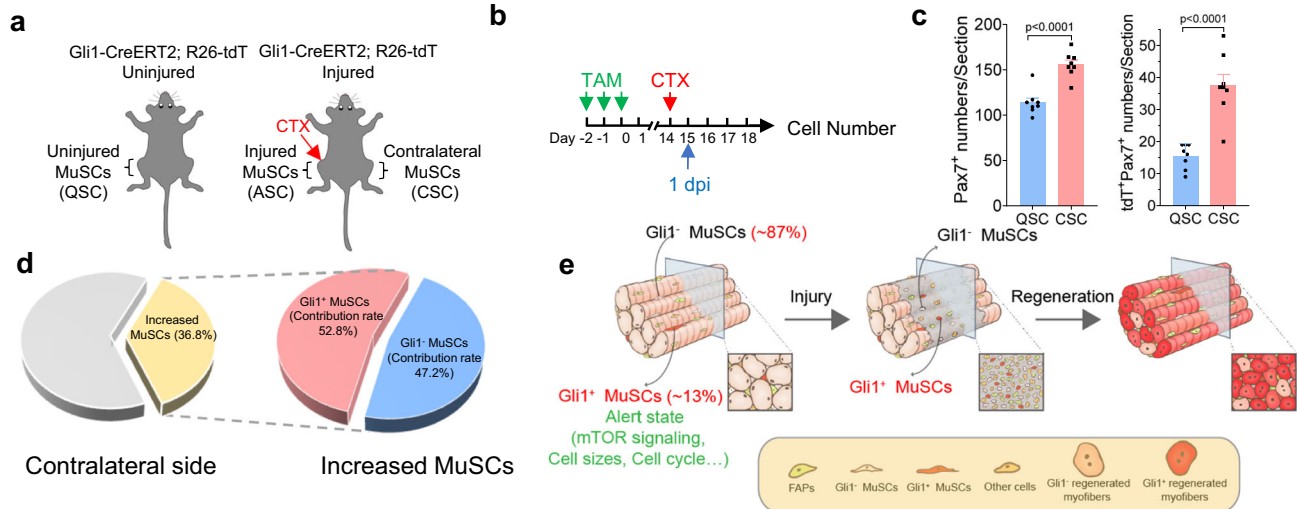

**Fig. 8 | Gli1⁺ MuSCs are sensitive to activation cues after injury. a** Scheme of the location of QSCs, CSCs and ASCs in relation to muscle injury. QSCs, quiescent MuSCs. CSCs, contralateral MuSCs. ASCs, activated MuSCs. **b** Scheme of the experimental strategy. **c** Statistical analysis of the number of total MuSCs or tdT⁺ (Gli1⁺) MuSCs per section in TA muscle of uninjured mice and the contralateral TA muscle of injured mice ($n = 8$). **d** Left panel: Quantification of the percentage of CSCs from the contralateral TA muscle of injured Gli1-CreERT2; R26-tdT mice at 1 dpi. The yellow histogram represented the percentage of increased MuSCs from the contralateral TA muscle compared with uninjured mice ($n = 8$). Right panel: Quantification of the percentage of tdT⁺ (Gli1⁺) and tdT⁻ (Gli1⁻) CSCs in increased MuSCs from the contralateral TA muscle. The blue histogram represented the percentage of increased tdT⁻ (Gli1⁻) CSCs. The red histogram represented the percentage of increased tdT⁺ (Gli1⁺) CSCs. dpi, day post injury. **e** Schematic overview of the participation of Gli1⁺ MuSCs in muscle regeneration. Gli1 marks a distinct population of MuSCs, which serves as sentinel cells to enable rapid response after injury and speed muscle regeneration. Data are presented as mean ± SEM; Statistical significance was determined by two-tailed unpaired Student's t test (**c**). All numbers ($n$) are biologically independent experiments. Source data are provided in the Source Data File.

expression in Gli1⁻ MuSCs was relatively high. All these results seem to suggest that Gli1⁺ MuSCs exhibit function and physiological properties, which is related to Gli3 low expression in uninjured muscle. These results indicated that in the uninjured muscle, there are a subset of MuSCs in a "shallower" quiescence status. These cells may serve as sentinels staying in alert to increase the sensitivity to injury and stress cues and participate regeneration immediately after injury. The existence of Gli1⁺ MuSCs speeds up the post injury response and ensure quick regeneration. They could be the cells frequently used to repair relatively minor injuries. It has been suspected that probably less than 15% of MuSCs contribute to the majority of the regeneration task[13]. Gli1⁺ MuSCs account for about 13% of MuSCs, while they contribute to over 50% of G$_{Alert}$ cells. These cells have stronger proliferation and differentiation ability and support muscle regeneration better than the Gli1⁻ MuSCs. Gli1⁺ MuSCs may be one of the major contributors for muscle regeneration. The Gli1⁻ MuSCs in deep quiescence may contribute more for the maintenance of stem cell pool. In the condition of severe injury, not only Gli1⁺ MuSCs, but also the reserved Gli1⁻ MuSCs contribute to muscle regeneration.

In conclusion, this study provides new insights into the homeostasis mechanism of the heterogeneous MuSC population. Our findings help us gain new insights on the dynamics of MuSC activation for tissue homeostasis maintenance. In the future, the matters of great interest will be identifying the important surface markers of Gli1⁺ MuSCs for better characterization of this important subset and studying its potential roles in human muscle regeneration. In addition, characterization of the cellular and molecular mechanisms of Gli1⁺ MuSCs in skeletal muscle regeneration will be interesting to allow us understanding better on the physiology of MuSC homeostasis and activation in muscle tissues.

## Methods
### Animals
Mice were housed and maintained in accordance with the guidelines of the Institutional Animal Care and Use Committee of the State Key

Laboratory of Cell Biology, Shanghai Institute of Biochemistry and Cell Biology, Center for Excellence in Molecular Cell Science, University of Chinese Academy of Sciences, Chinese Academy of Sciences. Gli1$^{tm2Alj}$/J (Gli1-LacZ), Gli1$^{tm3(cre/ERT2)Alj}$/J (Gli1-CreERT2), B6.129P2-Gt(ROSA) 26Sor$^{tm1(DTA)Lky}$/J (R26-DTA), Gt(ROSA)26Sor$^{tm1(rtTA,EGFP)Nagy}$ (R26-eGFP) and B6.Cg-Gt(ROSA)26Sor$^{tm9(CAG-tdTomato)Hze}$/J (R26-LSL-tdT) mice were purchased from Jackson Laboratory. C57BL/6, NOD/ShiLtJGpt-Prkdc$^{em26Cd52}$/Gpt (NOD-Scid) and B6/JGpt-Dmd$^{em1Cd647}$/Gpt (DMD-KO) mice were purchased from GemPharmatech. R26-RSR-tdT and R26-RSR-LSL-tdT (Ai66) mice were a gift from Dr. Bin Zhou. For Pax7-DreERT2, the cDNA encoding DreERT2 recombinase was inserted into the translation stop codon of the Pax7 gene. A P2A peptide sequence was used to link Pax7 coding region and DreERT2 cDNA, ensuring both transcriptions. The Pax7-DreERT2 line was generated by GemPharmatech. Gli1-CreERT2 and R26-LSL-tdT were crossed to generate Gli1-CreERT2; R26-tdT. Pax7-DreERT2 and R26-RSR-tdT were crossed to generate Pax7-DreERT2; R26-RSR-tdT. Gli1-CreERT2, Pax7-DreERT2 and Ai66 were crossed to generate Gli1-CreERT2; Ai66, Pax7-DreERT2; Ai66, Gli1-CreERT2; Pax7-DreERT2; Ai66. Gli1-CreERT2 and R26-DTA were crossed to generate Gli1-CreERT2; R26-DTA. Gli1-CreERT2; R26-eGFP/eGFP and Pax7-DreERT2; R26-RSR-tdT/tdT were crossed to generate Gli1-CreERT2; R26-eGFP; Pax7-DreERT2; R26-RSR-tdT. All mice were kept in group housing (2–5 mice/cage) in a specific pathogen-free (SPF) facility with controlled environmental conditions of temperature (20–25 °C), humidity (40–70%) and lighting (a 12 h light/dark cycle). All mice used for experiments were adults, between 8–12 weeks of age. The control and experimental mice used are littermates in all experiments. All mice used for experiments were kept at C57BL6/129 mixed background. Male and female mice were used in this study.

### Animal procedures
For lineage-tracing experiments, 8-week-old Gli1-CreERT2; R26-tdT, Pax7-DreERT2; R26-RSR-tdT, Gli1-CreERT2; Ai66, Pax7-DreERT2; Ai66, Gli1-CreERT2; Pax7-DreERT2; Ai66, Gli1-CreERT2; R26-DTA and Gli1-

CreERT2; R26-eGFP; Pax7-DreERT2; R26-RSR-tdT mice were injected intraperitoneally with 200 µg per gram body weight of 20 mg/ml Tamoxifen (TAM, dissolved in corn oil) daily for 3 consecutive days. Mice with intraperitoneal injection of corn oil were considered as vehicle control. Muscle injury studies were initiated in these animals following an over 2 weeks' washout period for TAM.

For muscle injury modeling, the tibialis anterior (TA) muscle was injected with cardiotoxin (CTX) (Sigma-Aldrich). Briefly, 50 µl of 10 µM CTX was injected at five injections site of TA muscles using 28 g needles. TA muscles were collected and analyzed at day 0, day 3, day 5, day 7 and day 14 after CTX injection.

For transplantation experiments, Gli1$^+$ MuSCs (tdT$^+$GFP$^+$) or Gli1$^-$ MuSCs (tdT$^+$GFP$^-$) were FACS isolated from Gli1-CreERT2; R26-eGFP; Pax7-DreERT2; R26-RSR-tdT mice at 3 days after TAM injection. Sorted MuSCs were injected into the TA muscle of Gli1-CreERT2; R26-DTA, DMD-KO or NOD-SCID mice.

## FACS and isolation of MuSCs

For isolation of MuSCs by FACS, after mice were euthanized, the hindlimb skeletal muscles were removed, chopped finely, and digested using 700 U/ml Collagenase type 2 (Worthington) in Wash medium (Ham's F-10 supplemented with 1% penicillin–streptomycin and 10% horse serum) for 60 min in shaking water bath at 37 °C. The digested muscle was washed twice with Wash medium and centrifuged at $500 \times g$ at 4 °C for 5 min. A second digestion was performed with 1000 U/ml Collagenase type II and 1.1 U/ml Dispase II in Wash medium for 30 min in shaking water bath at 37 °C. The twice-digested tissue was passed through a 20-gauge needle three times, then passed through a 70 µm filter and a 40 µm filter. The mononuclear muscle cells were stained using the following antibodies: FITC anti-mouse CD31, FITC anti-mouse CD45, FITC anti-mouse Sca1, APC anti-mouse CD106/VCAM1. Cells were incubated with primary antibodies for 60 min on ice, washed with cold Wash medium. DAPI were used for viable cell gating. FACS was performed using FACS Aria III (BD Biosciences) by gating for CD45$^-$CD31$^-$Sca1$^-$VCAM1$^+$ to isolate MuSCs.

For measurement cell size, tdT$^+$ (Gli1$^+$) and tdT$^-$ (Gli1$^-$) QSC, CSC, ASC were sorted based on CD45$^-$CD31$^-$Sca1$^-$VCAM1$^+$tdT$^+$ or CD45$^-$CD31$^-$Sca1$^-$VCAM1$^+$tdT$^-$. The cell size of tdT$^+$ (Gli1$^+$) and tdT$^-$ (Gli1−) QSC, CSC, ASC was measured by Coulter Multisizer 4e (Beckman).

For MitoTracker staining, the mononuclear muscle cells were washed and stained with the antibodies (FITC anti-mouse CD31, FITC anti-mouse CD45, FITC anti-mouse Sca1, PE/cy7 anti-mouse CD106/VCAM1) for gating the MuSC population. Then, about 40 nM MitoTracker Deep Red (Yeasen) were added to the mononuclear muscle cells and incubated for 30 min at 37 °C. The mitochondrial mass of tdT$^+$ (Gli1$^+$) and tdT$^-$ (Gli1$^-$) MuSCs was measured by FACS.

For puromycin incorporation assays, freshly isolated tdT$^+$ (Gli1$^+$) and tdT$^-$ (Gli1$^-$) MuSCs were treated with puromycin (10 µg/mL) for 15 min, lysed and processed for immuno-blotting.

## MuSC culture

FACS isolated MuSCs were either fixed in suspension immediately in 4% paraformaldehyde (PFA) for 10 min or maintained in collagen-coated dishes with Ham's F-10 supplemented with 20% FBS, 2.5 ng/ml bFGF and 1% Penicillin-Streptomycin, 50% T-cell conditional medium. Proliferating MuSCs were labeled by 5-Ethynyl-2′-Deoxyuridine (EdU) (5 ng/µl) for 12 h or 40 h at the end of the culture, and detected with the Click-iT EdU imaging kit (Invitrogen). Myogenic differentiation was induced with DMEM and 2% horse serum for 2 days. Fusion index was calculated as the number of nuclei in myotubes divided by the total number of nuclei counted.

## Single myofiber isolation and culture

Single myofibers were isolated from the extensor digitorum longus (EDL) muscle of 8-week-old mice following dissociation with collagenase I solution (0.2%) for 1.5 h at 37 °C. Dissociated single myofibers were manually collected and purified under a dissection microscope, then either fixed in suspension immediately in 4% PFA for 15 min, or maintained in suspension culture for 24, 48 and 72 h with plating media composed of Ham's F-10 supplemented with 1% Penicillin-Streptomycin, 20% FBS, 2.5 ng/ml bFGF and 1% Chicken Embryo Extract (CEE).

## Muscle X-Gal staining

For X-Gal staining, adult TA muscles were isolated and immediately frozen in OCT by liquid nitrogen. 10 µm sections were fixed for 15 min at 4 °C in LacZ fixative (100 mM MgCl$_2$, 0.2% glutaraldehyde, 5 mM ethylene glycol tetra-acetic acid in PBS). Fixed sections were rinsed in wash solution (0.1 M PBS containing 2 mM MgCl$_2$, 0.02% NP-40, 0.01% Na-deoxycholate, pH 7.4) three times and incubated in pre-warmed, filtered X-Gal staining solution (wash solution supplemented with 5 mM K$_3$Fe(CN)$_6$, 5 mM K$_4$Fe(CN)$_6$, and 1 mg/ml X-Gal) overnight in the dark at 37 °C. Prior to histological examination, some slides were counterstained with nuclear fast red staining solution following standard protocols.

## Immunofluorescence staining and imaging

Freshly isolated TA muscles were fixed in 4% PFA at 4 °C for 60 min. After fixation, tissues were washed in PBS three times, dehydrated in 30% sucrose overnight at 4 °C, embedded in OCT and flash frozen with liquid nitrogen. 10 µm sections were collected on glass slides and stored at −20 °C. For immunofluorescence staining, slides were put in a fume hood to air dry, followed by a brief PBS wash to remove OCT. Slides were blocked and permeabilized with 1% BSA and 0.1% Triton X-100 in PBS for 30 min, followed by incubation with primary antibodies overnight at 4 °C in the dark. Primary antibodies used in this study are listed in Supplementary Table 2. On the following day, the slides were washed in PBS for 10 min, 3 times to remove unbound primary antibodies, and then incubated with fluorescence-conjugated secondary antibodies for 60 min at room temperature in the dark. After washing in PBS for 5 min, 3 times to remove secondary antibodies, the sections were stained with DAPI (4′,6-diamidino-2-phenylindole) and mounted with Aqua-Poly/mount (Polysciences). For detection of Pax7 with anti-Pax7 (DSHB), a tyramide staining strategy was used. Cryosections were stained using TSA Kit per manufacturer's instructions (Akoya). For each experiment, the TA muscles were derived from at least 5 mice. Over 30 serial cryosections were collected, and the slices with maximum cross-sectional area were selected for immunofluorescence staining and quantification. For visualizing myofibers or MuSCs, myofibers or MuSCs were fixed using 4% PFA, blocked and permeabilized using 1% BSA and 0.1% Triton X-100 in PBS, and stained with primary antibodies against tdT, Pax7 or MyHC, then washed and stained with fluorescence-conjugated secondary antibodies. Counterstain of nuclei with DAPI was performed at the end of the staining (Invitrogen). Images were acquired on Olympus fluorescence microscope (BX53) or Leica TCS SP8 confocal microscope. ImageJ software was used to analyze the collected images.

## The CUBIC clearing

The CUBIC was performed as previously reported[45,46]. Briefly, freshly isolated EDL muscles from Gli1-CreERT2; R26-eGFP; Pax7-DreERT2; R26-RSR-tdT mice at 14 days after TAM injection were fixed in 4% PFA at 4 °C for 24 h, followed by wash with PBS. These EDL were immersed into 5 mL of 50% (v/v) CUBIC-L reagent (1: 1 mixture of water: CUBIC-L) for 1 day and further immersed in 5 mL of CUBIC-L reagent for 5 days. Next, these EDL were washed with PBS for 1 day and immersed in 5 mL

of CUBIC-R+ reagent for 4 days. EDL fluorescence images were acquired with Leica TCS SP8 confocal microscope.

### RNA extraction and real-time quantitative PCR
Total RNA was extracted from MuSCs using Trizol reagent (Invitrogen). Reverse transcription was performed using HiScript® III RT SuperMix for qPCR with gDNA wiper according to the manufacturers' instructions (Vazyme). Gene expression was assessed with AceQ Universal SYBR qPCR Master Mix (Vazyme) and analysis was performed using the $2^{-\Delta\Delta Ct}$ method. RT-qPCR were normalized to the housekeeping gene *Gapdh*. A list of primers is available in Supplementary Table 3.

### Western blotting
To detect Gli1, p-S6 and puromycin proteins in small numbers of cells, 20,000 sorted MuSCs Gli1+ or Gli1− MuSCs were lysed and diluted to protein concentration of 3 μg/μL using sample preparation kit (ProteinSimple). Cell lysates were examined using automated western blot system, WES System (ProteinSimple) as per manufacturer's protocol. Data analysis and quantitation of protein levels were performed using Compass Software (ProteinSimple). To detect Gli1, MyHC, MyoD and MyoG proteins in differentiated myotubes, differentiated tdT− and tdT+ myotubes were collected, centrifuged, and washed with 1× PBS. Whole myotubes were lysed in RIPA lysis buffer with 1× Protease Inhibitor Cocktail. Protein quantification was conducted using the BCA Protein Assay Kit. Next, Proteins were detected by western blot with the indicated antibodies. Primary antibodies used in this study are listed in Supplementary Table 2.

### RNA-sequencing
Total RNA was extracted from Gli1+ or Gli1− MuSCs using Trizol reagent (Invitrogen). Library construction was performed with 1 ng of input total RNA using the VAHTS® Universal V8 RNA-seq Library Prep Kit for Illumina (Vazyme). The libraries were sequenced with a NextSeq 500 (Illumina). RNA-seq reads were mapped to transcripts from Mus_-musculus.GRCm38 by STAR (star-2.7)[47]. Data were loaded into R using the tximport library and the gene/count matrix was filtered to retain only genes with five or more mapped reads in two or more samples. Differential expression was assessed using DESeq2. To further compare biological function of both Gli1+ and Gli1− MuSCs, Metascape (http://metascape.org/gp/index.html)[48] were performed.

### Single-cell RNA-sequencing
MuSCs or Gli1+ cells were enriched by FACS sorting into wash medium and centrifuged at $500 \times g$ at 4 °C for 5 min. Collected cells were resuspended in PBS + 0.5% BSA, then counted by hemocytometer and controlled for quality. If >98% of visible objects were verified to be single cells, the sorted cells passed quality control, and were processed with a Chromium Controller (10 x Genomics), captured and libraried with the Chromium Single Cell 3′ Library Kit v2 (10 x Genomics). cDNA libraries were sequenced on an Illumina Nextseq 500 (Novogene, Tianjin, China) to a minimum depth of 50,000 reads per cell. Aligned reads and gene-barcode matrices were then generated from FASTQ files including Read 1, Read 2 and i7 index using Cell Ranger (v.2.1.1) processing pipeline. Further analysis and visualization were performed with R package Seurat (v.3.1.1)[49,50]. Threshold of unique counts over 3500 or less than 200 was set to exclude cell doublets. Low-quality cells that have >6% mitochondrial counts were excluded. 'LogNormalize' method was conducted for normalization for each cell based on the total expression. 'FindVariableFeatures' was performed to detect variable genes across the single cells (selection.method = "vst"). The key parameters of 'FindClusters' were set with resolution = 0.05. The top markers were used to plot expression heatmap of marker genes. We re-grouped Gli1-expressing cells in Pax7+ MuSC populations. Briefly, cells with Gli1 gene expression greater than 0 were filtered out

and the gene expression data of these cells were extracted and converted into a data frame format. The Jensen-Shannon distance (JSD) between the expression of each gene and the Gli1 gene expression were calculated. Genes with JSD values more than 0.2 were saved as a CSV file. Clustering was performed with the top 55 genes with the lowest JSD values used as clustering markers.

### Statistics and reproducibility
All data were curated from multiple individual biological samples, and presented as mean ± SEM. For statistical analysis, unpaired two-sided Student's t-tests were performed for comparing differences between two groups, and One-way ANOVA tests for over two groups, using GraphPad Prism 8.0 software. Statistical significances were accepted when adjusted $p < 0.05$. The "n" in the study represented the number of biological replicates and was indicated in the manuscript. All mice were randomly assigned, and the investigators who analyzed the samples were blinded to the group allocations.

### Reporting summary
Further information on research design is available in the Nature Portfolio Reporting Summary linked to this article.

## Data availability
The Single-cell RNA-sequencing raw data generated in this study have been deposited in the National Center for Biotechnology Information (NCBI) Sequence Read Archive (SRA) database under BioProject accession number PRJNA1028159. The RNA-sequencing raw data generated in this study have been deposited in the NCBI database under accession code GSE239944. The GRCm38 data used in this study are available in the NCBI database under accession code GCF_000001635.20. Source data are provided with this paper.

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

## Acknowledgements

We thank the animal facility for animal husbandry, and technical supports from Core Facility for Cell Biology and the Genome Tagging Project Center at CEMC. We acknowledge Dr. Yingnan Lei and Xiangwei Wu (QianTang Biotech Co. Ltd) for the providing bioinformatics analysis and helpful comments. This study was supported by grants from the Strategic Priority Research Program of the National Key Research and Development Program of China (2020YFA0509003 to Y.Z.), the National Natural Science Foundation of China (32270787 to J.Y.P., 32130025 and 32293232 to Y.Z., 32170804 to P.H.), Shanghai Science and Technology Innovation Action (23ZR1469900 to J.Y.P.), Shanghai Municipal Science and Technology Major Project (Y.Z.) and Space Medical Experiment Project of China Manned Space Program (HYZHXM01017 to P.H.).

## Author contributions

Y.Z. and P.H. conceived and designed the experiments. J.Y.P., L.L.H. and B.L. performed most experiments. J.Y.P. and L.L.H. contributed to the computational analysis and statistical analysis. J.W.S, Y.A.W, K.P.W, Q.G., X.Y.L., Y.L., W.Q.W., S.L., X.F., C.L.Z., W.K.Z., J.J.Z. and S.B.S helped the experiments and provided technical support. J.Y.P., P.H. and Y.Z. wrote the manuscript.

## Competing interests

The authors declare no competing interests.
