## [Peer review file · Nature Communications]

REVIEWER COMMENTS

Reviewer #1 (Remarks to the Author):

The manuscript presented by Peng et al. use mouse models to mark and trace Gli1 expressing cells in an attempt to determine the function of these Gli1 expressing cells in skeletal muscle regeneration. The authors claim that there is a subpopulation of muscle stem cells that express Gli1 and are in a less quiescent state. The authors allege that these cells are more regenerative than the Gli1 negative MuSCs and contribute more to regeneration of injured muscle.

While the authors have numerous mouse models for lineage tracing and there is single cell data, they fail to perform the most basic of experiments and do not satisfactorily answer whether muscle stem cells even express Gli1 at the protein level, nor do they convincingly show the function of Gli1 in these cells.

Major concerns:

1. Not once did the authors stain MuSCs for Gli1 or perform a western blot for the protein.
2. In the single cell data, the expression level of Gli1 in the cell types is never shown, it would be important to show a umap for Gli1 expression in all cell types plus a violin plot of Gli1.
3. The RNA-Seq data is underutilized, the RPM level of Gli1, and the myogenic factors (Pax7, Myf5, Myod and Myogenin) should be shown. Additionally, it is unclear what threshold was used to determine the over 300 genes that are significantly differentially expressed. The one mention is for the volcano plot where a LFC of 0.5 is used, and no mention of the p-value threshold. Further, LFC 0.5 is rather loose, the standard used is LFC of 1. A PCA plot should be added. It is also not shown how many samples were sequenced. Finally, the accession number provided for the deposited sequencing data does not exist according to the site.
4. Claim that Gli1+ MuSCs are required/indispensable for muscle regeneration, but that is a vast overstatement and no experiment in this manuscript answers that. If anything, the fact that there are tdT- regenerating myofibers would go against that claim and would mean that Gli1- MuSCs are sufficient for regeneration.
5. Claim that Gli1- MuSCs do not become Gli1+. However, the experiment used to determine this treats cultured myoblasts with tamoxifen to see if the tdT- cells eventually become tdT+. The issue here is that myoblasts do not express Gli1, therefore they can never become tdT+. A more appropriate experiment that the authors must perform is to transplant tdT- cells (and by extension Gli1- cells) into a host, either NOD Scid or DMD mouse and then treat the host mice with tamoxifen. If the host mice gains tdT+ cells that would indicate that the Gli1- cells can become Gli1+. The opposite would also be important to show, do the Gli1+ cells become Gli1-, the same experiment described in the previous sentence should be performed, this time transplanting tdT+ cells and then staining the tdT+ cells for Gli1. If there are Gli1-/tdT+ cells that would indicate that MuSCs that express Gli1 can convert into the negative population. This would have implications for the experiments where it is shown that tdT+ populations grow more quickly than the tdT- as those experiments only show that the original cell may have expressed Gli1, not that the entirety of the population remains Gli1+
6. An important experiment would be to isolate myofibers from mice that had been treated with tamoxifen. Fix the myofibers, 0, 24, 48 and 72 hours after isolation and stain for Pax7, tdT, Myogenin, and Gli1 to analyze how the Gli1+ cells behave. Are the tdT+ cells expanding more than the tdT-, do the cells continuously express Gli1 or is its expression transient and is there a difference in the proportion of Pax7+ and Myog+ cells between the tdT+ and tdT- cell populations. This experiment will effectively show whether Gli1 is expressed in quiescence, and whether it continues to be expressed throughout activation and proliferation. Further, it would confirm the authors' claims that Gli1+ cells are more proliferative.
7. Through transplantation it is said that the Gli1+ cells contribute more to regeneration than the Gli- cells. However, there is no way to track the Gli1- cells, and as such there is no way to determine directly their overall contribution.

8. In Figure 6 panel L and M and the text line 343, the authors say that over 50% of MuSCs in the contra lateral leg of an injured mouse were Gli1+, in comparison to uninjured mice which is 13%. However, when looking at the data even a cursory glance shows that the number of Gli1+ cells in the CSC condition is nowhere near 50% despite the claims. In Figure 6 panel L, the total number of MuSCs/section is about 150 according to the graph, and the total number of Gli1+ cells in the CSC is generously 40 cells, again according to the graph, therefore the percentage of Gli1+ MuSCs is at most 26.67% ($40/150 * 100 = 26.67$) which is only half of the number the authors are reporting in the text and in Figure 6 panel M.

9. It is not clear if the Gli1+ cells which are a very small percentage are contaminants or that their presence in scRNA-seq is due to contaminating ambient RNA. The authors could also run RNA velocity on the single cell data to see if their so called Gli1+ cells originate from Gli1- or vice versa.

Minor concerns:

1. Could the authors clarify which Vcam1 antibody was used for which experiments
2. The authors should include a control experiment where after FACS sorting tdT+ cells to stain them for Pax7, to confirm the percentage of sorted cells that are actually MuSCs
3. Line 140-142 says "Immunofluorescence staining and FACS analysis on TA muscle sections and MuSCs showed that over 85% of tdT+ (Pax7+) cells were expressing Pax7...". The way this is phrased indicates that only 85% of the tdT+ cells are Pax7+, yet in the supplemental figure 2 the graph says that 85% of Pax7+ cells are tdT+, which one is correct?
4. Figure 2 Panel E is not very representative of the data presented
5. Figure 2 Panel F, the overlay panel is brighter than the individual channels
6. To expand on the reported increase in differentiation, a western blot of differentiated tdT- and tdT+ myotubes for Gli1, MyHC, Myod, and myogenin would be informative
7. Figure 5 Panel I, legend says n=6-10 mice used, however in the tdT- cells there are 10 data points in time to exhaustion and 11 data points in distance traveled. Further, the tdT+ cells have 11 data points.
8. Figure 5 Panel E, binning the CSA would be more informative
9. Figure 5 Panel H, add the weight of the uninjured TA as an extra control
10. Figure 6 Panel B, the staining is very unconvincing. The Pax7 stain looks to be background
11. In the methods section, it is mentioned that isolated fibers were either fixed immediately after isolation or were cultured for 48 hours. However, there are no experiments performed on cultured myofibers

Reviewer #2 (Remarks to the Author):

Skeletal muscle is capable of robust regeneration thanks in part due to a rare population of skeletal muscle stem cells (MuSCs), also known as satellite cells for their satellite position to the myofibre, and underneath the basal lamina. Recent studies demonstrate substantial heterogeneity of MuSCs at various levels, including expression of transcription factors and surface markers, clonal ability, transplantation efficacy and functional responses to environmental stress. It is still unclear to what extent MuSCs are heterogeneous and what the functional implications of MuSC heterogeneity may be.

This is an interesting manuscript that has described a subset of MuSCs that uniquely express the GLI family transcription factor GLI1. Using genetic tools in the mouse, the authors show that approximately 15% of MuSCs express GLI1, depletion of the GLI1-expressing (GLI1(+)) MuSC population leads to delayed skeletal muscle regeneration, and that GLI1(+) MuSCs transplant more efficiently than GLI1(-) MuSCs. The authors further characterize GLI1(+) MuSCs as 'primed', characterized by increased mTORC1 signaling, increased cell size and mitochondrial numbers, reminiscent of a recently characterized 'Galert' phenotype of MuSCs that exist in skeletal muscle contralateral to acute injury. The authors conclude that GLI1(+) MuSCs exist as a sentinel population

of MuSCs that rapidly respond to muscle injury.

There are many strengths to the manuscript as follows:

1. Elegant use of multiple genetic mouse models convincingly support the authors conclusions of a GLI1(+) MuSC population, and that this population of MuSCs contribute to MuSC regeneration.
2. To my knowledge the GLI1 member of the GLI family transcription factors has not been studied in MuSCs, although additional members (GLI3 and GLI2) have recently been shown to regulate quiescence and activation, respectively.
3. The notion of a 'sentinel' population of MuSCs (in the absence of acute damage elsewhere) is interesting and new.
4. In general, the authors conclusions are supported by high quality, convincing data throughout (but with some exceptions stated below).

There are however a number of important weaknesses that the authors need to address to support their conclusions:

1. Very recently, two papers have been published establishing cilia as important regulators of MuSC activity. Primary cilia function as hubs of hedgehog signaling, the effects of which are mediated by the GLI family of transcription factors. GLI3 is required for maintaining MuSC quiescence, and MuSCs lacking GLI3 enter the Galert state in the absence of injury via activation of mTORC1 signaling (PMID: 35803939). GLI2 is transiently upregulated upon MuSC activation, and aging MuSCs are defective for both primary cilium and GLI2 expression at basal state and after acute injury (PMID: 35301320). In light of these recent contributions, a question of novelty of the present work arises. In my opinion, the current manuscript has a strong foundation to significantly contribute new knowledge, but needs to address the following recommendations:

a. The authors have cited both of the papers I highlight above, but in my opinion these citations come too late in the manuscript. The Introduction of the manuscript should include a stronger introduction to the GLI family of TFs (page 4, last paragraph), and cite the current knowledge about cilia, HH signaling and GLI family of transcription factors in muscle stem cells and in myogenesis. An introduction providing the current knowledge will put the authors in a stronger position to highlight their new questions and novel contributions.

b. To my knowledge, GLI1 is primarily a transcriptional activator, while GLI3 is a transcriptional repressor or activator. Since the absence of GLI3 was shown to cause MuSCs to adopt a Galert state (PMID: 35803939), I would like to know Gli3 expression levels in the subset of GLI1(+) MuSCs. If GLI1(+) MuSCs do express Gli3, how do the authors reconcile?

2. I would like to see the GLI1(+) MuSCs characterized in greater detail with respect to their phenotype as follows:

a. Given the link to mTORC1 signaling and Galert, what are levels of protein synthesis in GLI1(+) vs GLI1(-) MuSCs, potentially addressed by puromycin incorporation assays.

b. What is the relative expression of Pax7/PAX7, CD34, Myf5/MYF5 and MyoD/MYOD in GLI1(+) vs GLI1(-) MuSCs, potentially addressed with the authors existing sequencing datasets for transcript levels and by western blotting or immunofluorescence with antibodies against PAX7, CD34, MYF5, MYOD.

c. What is the relative ability of GLI1(+) MuSCs vs GLI1(-) MuSCs to contribute to self-renewal, a

defining property of stem cells. This is potentially addressed in vivo by transplantation assays, followed by donor cell contribution to the PAX7(+) MuSC pool. In my opinion, this additional knowledge is essential to support the authors conclusions that GLI1(+) MuSCs have enhanced regeneration after acute injury and after engraftment.

3. Related to Figure 3G, the authors conclude that GLI1(+) MuSCs have enhanced proliferation and enhanced differentiation. However, these two processes are mutually exclusive given that differentiating MuSCs exit the cell cycle to initiate fusion.

a. Given the experimental setup (2days proliferation medium, 2 days differentiation medium), it is likely that enhanced proliferation in the first two days gives rise to more myogenic progenitors that can give rise to a greater fusion index.

b. If GLI1(+) MuSCs have enhanced differentiation, one might expect they precociously differentiate at the expense of proliferation. Precocious differentiation is potentially addressed using antibodies against Myogenin at early timepoints prior to fusion.

c. If the authors want to demonstrate that GLI1(+) MuSCs have enhanced differentiation, the starting cell density should be the same at the step of changing to differentiation media.

4. A lot of data presented in Figure 5 are problematic for the following reasons:

a. The main problem being the Gli1(-) population cannot be tracked due to the lack of a fluorescent reporter. Although the data shows GLI1(+) cells contribute to regeneration after engraftment, strong conclusions about the efficiency of GLI1(+) vs GLI1(-) population cannot be made.

b. Figure 5d-e, multiple timepoints after engraftment would help. Additional markers for embryonic myosin heavy chain (embMHC; regenerating myofibres), PAX7 (donor MuSC population) would give more informative results.

c. Essential to include non-engrafted control to observe the collapse of regeneration following DTA, and also to prove the DTA is working as expected to eliminate PAX7(+); GLI1(+) MuSCs.

d. Notably the DTA experiments will eliminate this minor population of GLI1(+) MuSCs, but also essential supporting cells such as GLI1(+) FAPs, making interpretation of the results very difficult.

e. Figure 5c-e. I suppose the Gli1-CreERT2 will leave about 85% of the MuSC population intact, making interpretation of results difficult.

f. Essential to include non-engrafted control to observe the collapse of regeneration following irradiation, and also to prove the irradiation has eliminated the host MuSC population.

g. Figure 5g. I would normally insist to please report the number of donor derived myofibres as a percentage of total myofibres, but again the GLI1(-) population cannot be tracked. The efficiency of engraftment is important to understand Figure 5i.

h. The functional recovery reported in Figure 5i is remarkable given the experimental setup. It's essential to have appropriate controls, such as the PBS control in Figure 5i.

5. Engraftment of donor cells into the Dmdmdx mouse model confers some advantages in that both GLI1(+) and GLI1(-) engraftment efficiency can be reported by numbers of donor derived myofibres expressing dystrophin. However, the dystrophin immunolabeling in Supplementary Figure 5h lower panels needs to be improved. The lower panels should look like the upper panels with respect to dystrophin, and the presented data raises questions whether the number of dystrophin(+) myofibres counted is accurate. The results should also show whether 18Gy irradiation eliminated endogenous

MuSCs.

Minor points that don't require experimental revisions:

1. Pg 10 line 203 and Supplementary Figure 6 title. The data presented does not support the conclusion that GLI1(+) MuSCs are 'indispensible' for regeneration. The Figure 5 title is more appropriate heading given the results presented.
2. Figure legends for Figure 5b-c are not correct, I think c refers to b and vice versa.
3. To increase readability, the authors could consider supplementing immunofluorescence images with short descriptors of experimental design (eg/ timepoints after ctx injury, age of mice, donor MuSC identity) underneath the images.
4. Figure 3h, please explain the Fusion index in the y-axis (what are the results a percentage of?)

Reviewer #3 (Remarks to the Author):

Summary

In this manuscript, Peng et al. identified a novel subpopulation of muscle stem cells marked by the expression of Gli1. By scRNA-seq on skeletal muscle isolated from a conditional and inducible mouse model, in which all Gli1-expressing cells are labeled by the tdTomato (tdT) upon tamoxifen induction (Gli1+ tdT+), along with immunostaining on muscle sections and isolated myofibers, the authors show that around 10% of muscle stem cells (MuSCs) are expressing Gli1. Analyzing a cleverly designed mouse model that allows tracking Gli1 expression in muscle stem cells specifically (Pax7+ Gli1+ tdT+), as well as the Gli1-lacZ mouse line, further confirms their results. To characterize the Gli1+ cell population, the authors perform bulk RNA-seq analysis and in vitro assays on Gli1- and Gli1+ MuSCs and show that Gli1+ MuSCs exhibit enhanced proliferation and differentiation capacity. Then, they analyze the regenerative capacity of the Gli1+ MuSCs through a series of in vivo experiments. Cardiotoxin-induced muscle injury reveals that Gli1+ MuSCs participate to muscle regeneration and depleting this cell population alters the regeneration process. Transplantation experiments to show that Gli1+ MuSCs exhibit a higher ability to regenerate. Finally, the authors show that Gli1+ MuSCs mimic the GAlert state, conferring them a faster ability to activate and regenerate.

Critiques/Comments/Suggestions

1. The authors claim the existence of a subpopulation of Gli1-expressing muscle stem cells. However, the scRNA-seq data do not identify a distinct cluster for Gli1-expressing cells from FACS-isolated CD45- CD31- Sca1- VCAM+ cells (Fig. 1g-i). Instead, they show that some cells within the muscle stem cell population are expressing Gli1. Thus, the term of 'subpopulation' should be modified. To this reviewer's point of view, Gli1 maybe marks a state of quiescence but does not define a subpopulation of MuSCs. Also, from the Fig. 1g, would it be possible to get the proportion of muscle stem cells that are expressing Gli1? Does it correlate with the proportions of Gli1+ tdT+ MuSCs observed in Fig. 2?
2. The manuscript lacks mechanistic studies that link Gli1 expression to the cellular phenotypes of the Gli1+ MuSCs.
 - a. Although the bulk RNA-seq analysis of the Gli1/tdT+ MuSCs shows the upregulation of downstream mediators of Hedgehog signaling that might regulate their cellular phenotypes (Fig. 3c, Gli1, Ptch1, Hhip), none of them are explored in the study. What is the expression of Gli1 in quiescent and activated muscle stem cells? In cultured myoblasts and myotubes? How does the Hedgehog signaling

behave during MuSC progression in the myogenic lineage? Is Hedgehog signaling involved in the cellular phenotypes of Gli1+ MuSCs?

b. Besides cell proliferation and differentiation, the transcriptome analysis on freshly FACS-isolated Gli1/tdT+ and Gli1/tdT- muscle stem cells also reveals gene ontology terms related to regulation of IGF transport and mTOR signaling (Fig. 2c, d). How do the authors link Gli1 to these signaling pathways? Does it stand true for the cultured MuSCs (myoblasts) and differentiated cells (myotubes)? Are the down-/up-regulated genes depicted in Fig. 2e still differentially expressed in cultured muscle cells? Especially, are there any differences in the Insulin-like growth factor and mTOR signaling activation? For instance, mTOR signaling is required for myogenic progenitor proliferation and fusion (10.1242/dev.172460; 10.1016/j.bbrc.2015.05.032). Increased mTOR activity in Gli1+ muscle cells would not only explain the GAlert phenotype in MuSCs but also the increased ability of the Gli1+ myogenic progenitors to proliferate and differentiate. The authors should assess the mTOR signaling activity in Gli1- and Gli1+ myogenic cell culture.

3. As mentioned in the introduction, the Hedgehog signaling effector Gli1 is expressed in stromal cells of many organs but it remains unclear whether Gli1 expression marks a subset of stem cells or progenitors. Using the tdT reporter to trace the Gli1+ muscle stem cell population, the authors show that Gli1+ MuSCs participate to muscle regeneration as around 75% of the newly formed myofibers are labeled by the tdT (Fig. 4). However, they do not assess whether the Gli1+ tdT+ cells can self-renew.

a. As they are more prone to proliferate, differentiate and fuse (Fig. 3), one could think that this cell population resemble more a committed myogenic progenitor population that will be dedicated for rapid regeneration upon an injury, rather than a muscle stem cell population that will self-renew and repopulate the niche following the injury. To tackle this question, the authors should quantify the number of tdT- and tdT+ PAX7-expressing cells on cross-sections of regenerated muscles (Fig. 4).

b. PAX7-expressing cells should be also quantified on cross-sections of recipient muscle following the tdT- and tdT+ MuSC transplantation (Fig. 5). That way, the authors will determine whether the tdT+ cells represent a population of stem cells or a population of committed progenitor cells.

c. Although they perform flow cytometry analysis to show that there is still a proportion of CD45- CD31- Sca1- VCAM+ mononucleated cells that express tdT, they cannot conclude if this population maintains PAX7 expression (VCAM marks both the PAX7+ muscle stem cells and the differentiated Myogenin+ muscle cells). Again, quantifying the number of tdT- and tdT+ PAX7-expressing cells would be much better to characterize each cell type.

4. Regarding the transplant experiments (Fig. 5), a major concern is that there is no means to track the tdT- muscle stem cell population in order to compare their engraftment capability with the tdT+ MuSCs. Therefore, it is difficult to conclude anything on the comparison made.

a. This reviewer would suggest either to use a double-fluorescent cre-reporter mouse (such as the mT/mG) instead of the tdT reporter mouse or to cross their mice (Gli1-CreERT2; Pax7-DreERT2; Ai66) with the CAG-GFP transgenic mice. Thus, the authors will track both the Gli1- (mT+ or GFP+) and Gli1+ (mG+ or GFP+ tdT+) cells following their transplantation.

b. In Fig. 5g, the muscle picture of the NOD-SCID recipient mouse is not injured (no centrally located nuclei, no sign of regeneration) while it should be. Yet tdT+ myofibers are observed. This observation supports the idea that Gli1+ tdT+ cells are differentiating faster compared to the tdT- and are pro-fusogenic. Please revise your figure. As well and for each engraftment experiment, please provide a representative picture of the full muscle section.

c. The authors transplanted the tdT- and tdT+ MuSCs within irradiated and injured muscle of mdx mice. That way, they can track the fate of the engrafted MuSCs without relying on a reporter (Supp Fig. 6). These panels should probably appear in the main figures as it supports their conclusion. However, as mentioned above, quantifying the number of tdT- and tdT+ PAX7-expressing cells is required to conclude on the status of each cell type. As well, can the author add a Laminin immunostaining to show the surrounding mdx myofibers? Also, please provide a better staining for dystrophin in the tdT+ MuSC condition.

5. The authors only quantify the proportions of the tdT⁻ and tdT⁺ cell populations in Supp. Fig. 4d, Fig. 6h and Supp. Fig. 7c. Thus, it looks like the tdT⁺ cells are increasing during the first few days of regeneration while the tdT⁻ are decreasing. One could expect that both cell types are increasing in numbers but the tdT⁺ proliferate faster than the tdT⁻. Providing the absolute numbers for each cell types would clarify their results and conclusion.

6. Figure 2f: Would it be possible to get quantify the number of PAX7⁺ tdT⁻ and PAX7⁺ tdT⁺ muscle stem cells per myofiber?

7. Lines 206-212: This paragraph should appear earlier in the 'results' section. Given that CreERT2 insertion in the Gli1 allele can affect Gli1 expression, it seems important to mention that the Gli1-CreERT2 allele is kept heterozygous throughout the study, especially for the in vivo Gli1⁺ tdT⁺ cell tracing and regeneration experiments.

8. Lines 336-338: "tdT⁺ (Gli1⁺) MuSCs display rapid cell-cycle entry, increased proliferation and augmented self-renewal and markedly enhanced regenerative capacity in the contralateral TA muscle (Supplementary Fig. 7e-i)". This is a very strong statement. The authors did not assess the self-renewal ability of the tdT⁺ cells. Moreover, they cannot assess their regenerative capacity as the experiment was performed at day 1 post-injury. Please modify/clarify.

9. a. Figure 6b: There is no PAX7 staining on the representative image selected. Given that the tdT labels all the Gli1-expressing cells independently of Pax7 expression, the cell shown in the image might be a PAX7⁻ cell, such as a FAP that can also transit from G0 to GAlert (doi:

10.1038/nature13255). To avoid any confusion, the PAX7 staining should be redone/improved. As well, the authors could examine the expression of phospho-S6 in satellite cells from freshly isolated myofibers and quantify the number of pS6⁺ tdT⁺ and pS6⁺ tdT⁻ MuSCs.

b. Figure 6e: There is only one replicate of each genotype shown and therefore, no quantification of the mitochondrial mass for each genotype. Quantifying the mean fluorescence intensity for 3 biological replicates would strengthen the data.

c. Figure 6i: If Gli1 expression is not maintained following MuSC commitment into the myogenic lineage, it is not surprising that the Gli1⁻ myoblasts do not turn on the tdT upon 4-hydroxytamoxifen treatment in vitro. Evaluating Gli1 expression seems to be important for the study (see comment 2a).

d. Figure 6l: It would be nice to see representative images for all the quantifications shown. That way, the data could be more appreciated.

e. Figure 6m: Legends are unclear. Shouldn't it be written uninjured and contralateral side instead of MuSCs and increased MuSCs?

Point-by-point response to Reviewers' comments and concerns

We would like to thank all the reviewers for their careful assessment of our manuscript, which we believe has helped us significantly to improve the quality of our work. We have now taken all comments into account and presented a revised manuscript, providing new data and analyses in response to the reviewer's comments. In addition, revised portions are marked in red in the revised manuscript according to requirements of this journal. A point-by-point response is set out below. We hope that these responses will satisfy the reviewers. Thanks!

Reviewer #1 (Remarks to the Author):

The manuscript presented by Peng et al. uses mouse models to mark and trace Gli1-expressing cells in an attempt to determine the function of these Gli1-expressing cells in skeletal muscle regeneration. The authors claim that there is a subpopulation of muscle stem cells that express Gli1 and are in a less quiescent state. The authors allege that these cells are more regenerative than the Gli1-negative MuSCs and contribute more to regeneration of injured muscle.

While the authors have numerous mouse models for lineage tracing and there is single-cell data, they fail to perform the most basic of experiments and do not satisfactorily answer whether muscle stem cells even express Gli1 at the protein level, nor do they convincingly show the function of Gli1 in these cells.

Response: We agree with the reviewer that it's very important to explicitly show the expression level of Gli1 protein in MuSCs. The expression of Gli1 was detected in the single-cell RNA sequencing and RNA sequencing from MuSCs as shown in **revised Fig. 1i** in the revised manuscript. We now also performed immunofluorescence staining on freshly isolated tdT⁻ and tdT⁺ MuSCs from Gli1-CreERT2; R26-tdT mice. The expression of Gli1 protein can be detected in tdT⁺ MuSCs (**revised Fig. 3b**). Next, we pooled tdT⁺ MuSCs from 10 Gli1-CreERT2; R26-tdT mice to obtain cells sufficient for Western blot. The same number of Gli1⁻ MuSCs were also subjected for Western blot analysis. Gli1 protein also can be detected in tdT⁺ MuSCs (**revised Fig. 3d**). Additionally, freshly isolated myofibers (0 h) and cultured myofibers (24 h, 48 h and 72 h) from Gli1-CreERT2; R26-tdT mice were then fixed in 4% PFA and stained for Gli1. Gli1 is exclusively expressed in tdT⁺

MuSCs of myofibers (**revised Supplementary Fig. 7g**). Taken together, these results further confirmed the expression of Gli1 in MuSCs.

The expression of Gli1 in MuSCs raises an interesting question about its role. Previous studies have demonstrated the pivotal role of the Hedgehog (Hh) signaling pathway in muscle regeneration, involving in both canonical and non-canonical mechanisms (Palla et al, 2022, PMID: 35301320; Brun et al, 2022, PMID: 35803939). Gli family of transcription factor play important roles as Hh signaling effectors. Recent study found that Gli3 is required for maintaining MuSCs quiescence, and MuSCs lacking Gli3 enter the G_{Alert} state in the absence of injury via activation of mTORC1 signaling (Brun et al, 2022, PMID: 35803939). Gli2 is transiently upregulated upon MuSCs activation, and aging MuSCs are defective for both primary cilium and Gli2 expression at basal state and after acute injury (Palla et al, 2022, PMID: 35301320). Gli1, as a direct target gene, plays a role in positive feedback by enhancing Gli activity. Thereby, we have reason to suspect that Gli1 may play an important role in muscle regeneration. However, in this study, we mainly focus on the effects of Gli1+ MuSCs on muscle regeneration. In future work, we will elucidate the precise role of Gli1 in MuSCs, as it might provide further interesting insight.

Major concerns:

1. Not once did the authors stain MuSCs for Gli1 or perform a western blot for the protein.

Response: We thank the reviewer for the suggestion. As described above, we performed Gli1 immunofluorescence staining and Western blot analysis as suggested by the reviewer (**revised Fig. 3b, d, revised Supplementary Fig. 7g**). These results confirmed the expression of Gli1 in MuSCs.

2. In the single cell data, the expression level of Gli1 in the cell types is never shown, it would be important to show a umap for Gli1 expression in all cell types plus a violin plot of Gli1.

Response: We thank the reviewer for the suggestion. Following this suggestion, we have generated a t-SNE graph showing the expression of *Gli1* in all cell types detected by the single cell sequencing analysis from skeletal muscle (**revised Supplementary Fig. 1b**).

3. The RNA-Seq data is underutilized, the RPM level of Gli1, and the myogenic factors (Pax7, Myf5, Myod and Myogenin) should be shown. Additionally, it is unclear what threshold was used to determine the over 300 genes that are significantly differentially expressed. The one mention is for the volcano plot where a LFC of 0.5 is used, and no

mention of the p-value threshold. Further, LFC 0.5 is rather loose, the standard used is LFC of 1. A PCA plot should be added. It is also not shown how many samples were sequenced. Finally, the accession number provided for the deposited sequencing data does not exist according to the site.

Response: We thank the reviewer for the suggestion. We apologize for the confusion about the accessing number of the sequencing data. We also resubmitted the RNA-seq data to NCBI. The accession number is GSE239944. We have double checked the accession number and made the data openly accessible. We now used LFC of 1, p value < 0.01 to reanalyze our sequencing data. 208 genes are significantly differentially expressed (**revised Fig. 3e**). The FPKM values of Gli1, Pax7, Myf5, MyoD and Myogenin also have been graphed in **revised Supplementary Fig. 6c, d**. As suggested by the reviewer, PCA analysis also was performed (**revised Supplementary Fig. 6b**). The expression profile of Gli1+ MuSCs showed significant difference from Gli1- MuSCs.

4. Claim that Gli1+ MuSCs are required/indispensable for muscle regeneration, but that is a vast overstatement and no experiment in this manuscript answers that. If anything, the fact that there are tdT- regenerating myofibers would go against that claim and would mean that Gli1- MuSCs are sufficient for regeneration.

Response: We thank the reviewer for the suggestion. We have revised the manuscript accordingly to make our description and conclusion more accurate. We have changed the conclusion as “Gli1+ MuSCs are critical for proper muscle regeneration.” in the revised manuscript.

5. Claim that Gli1- MuSCs do not become Gli1+. However, the experiment used to determine this treats cultured myoblasts with tamoxifen to see if the tdT- cells eventually become tdT+. The issue here is that myoblasts do not express Gli1, therefore they can never become tdT+. A more appropriate experiment that the authors must perform is to transplant tdT- cells (and by extension Gli1- cells) into a host, either NOD Scid or DMD mouse and then treat the host mice with tamoxifen. If the host mice gains tdT+ cells that would indicate that the Gli1- cells can become Gli1+. The opposite would also be important to show, do the Gli1+ cells become Gli1-, the same experiment described in the previous sentence should be performed, this time transplanting tdT+ cells and then staining the tdT+ cells for Gli1. If there are Gli1-/tdT+ cells that would indicate that MuSCs that express Gli1 can convert into the negative population. This would have implications for the experiments where it is shown that tdT+ populations grow more quickly than the tdT- as

those experiments only show that the original cell may have expressed Gli1, not that the entirety of the population remains Gli1+

Response: We thank reviewer for the suggestion. Following the reviewer's suggestions, we performed transplantation experiments on Nod-Scid mice. TAM was injected intraperitoneally on the second day after tdT- (Gli1-) MuSCs transplanting (**revised Supplementary Fig. 10a**). After TAM treatment, we did not observe tdT⁺ fibers or cells in the recipient mice, indicating that tdT- (Gli1-) MuSCs do not convert to tdT⁺ (Gli1+) MuSCs *in vivo* (**revised Supplementary Fig. 10b**). Next, Gli1+ MuSCs were transplanted into the pre-irradiated TA muscles of Nod-Scid mice 1 day after CTX-induced injury (**Referee Fig. 1a**). TA muscles were harvested 30 days after transplantation, and the expression of Gli1 was examined by immunofluorescence staining. 30 days after the transplantation, as expected, many myofibers were also tdT⁺ (**Referee Fig. 1b**), suggesting that the transplanted tdT⁺ (Gli1+) MuSCs participate the muscle regeneration. We also observed Gli1 expression in almost all of tdT⁺Pax7⁺ MuSCs, suggesting that some of the transplanted MuSCs retain Gli1 expression (**Referee Fig. 1b**). In addition to the Gli1+tdT⁺ MuSCs, population of tdT⁺ cells or tdT⁺ myofibers emerges that no longer expresses endogenous Gli1 protein. These results confirmed that tdT⁺ (Gli1+) MuSCs may convert to tdT- (Gli1-) cells *in vivo*. However, further study is needed to determine whether Gli1 expression is necessary for the faster growth of tdT⁺ populations compared to tdT-

Referee Fig. 1 **a** Scheme of the experimental strategy. **b** Particular numbers (5,000 cells) of freshly isolated tdT⁺ (Gli1+) MuSCs from Gli1-CreERT2; R26-tdT mice were transplanted into pre-injured TA muscle of the NOD-SCID mice. TA muscles were harvested 30 days after transplantation. Scale bars, 50 μ m.

populations.

6. An important experiment would be to isolate myofibers from mice that had been treated with tamoxifen. Fix the myofibers, 0, 24, 48 and 72 hours after isolation and stain for Pax7, tdT, Myogenin, and Gli1 to analyze how the Gli1+ cells behave. Are the tdT+ cells expanding more than the tdT-, do the cells continuously express Gli1 or is its expression transient and is there a difference in the proportion of Pax7+ and Myog+ cells between the tdT+ and tdT- cell populations. This experiment will effectively show whether Gli1 is expressed in quiescence, and whether it continues to be expressed throughout activation and proliferation. Further, it would confirm the authors' claims that Gli1+ cells are more proliferative.

Response: We thank the reviewer for the valuable suggestion. As suggested by the reviewer, we isolated myofibers from Gli1-creERT2; R26-tdT mice, and then fixed them at 0 h, 24 h, 48 h and 72 h after *in vitro* culture. Immunofluorescence staining of Pax7 revealed that both Gli1+ and Gli1- MuSCs expressed Pax7 at all-time points (**revised Supplementary Fig. 7f**). Furthermore, we performed immunofluorescence staining of Gli1 and MyoG on fibers at each time point. The results showed that only tdT+ MuSCs expressed Gli1 and maintained its expression from 0 h to 72 h (**revised Supplementary Fig. 7g**). Additionally, we observed only weak MyoG signals in the 72 h samples (**revised Supplementary Fig. 7h**). Upon analyzing fibers at 72 h, we observed a significantly higher number of Gli1+ MuSCs per muscle fiber compared to Gli1- MuSCs (**revised Supplementary Fig. 7i**). The above experiment results confirmed that Gli1+ MuSCs are more proliferative, which is consistent with our results from cell culture experiments *in vitro*.

7. Through transplantation it is said that the Gli1+ cells contribute more to regeneration than the Gli- cells. However, there is no way to track the Gli1- cells, and as such there is no way to determine directly their overall contribution.

Response: We thank the reviewer for pointing out this important issue. We agree with the reviewer that tracking both Gli1+ and Gli1- MuSCs after transplantation will greatly strengthen the notion that Gli1+ MuSCs support muscle generation better. In order to track both Gli1- and Gli1+ MuSCs, we generated a new mouse strain (Gli1-CreERT2; R26-eGFP; Pax7-DreERT2; R26-RSR-tdT) using a new genetic lineage tracing system that utilizes Dre and Cre recombinases to examine Gli1- and Gli1+ MuSCs. As shown in **revised Supplementary Fig. 8d**, Gli1+ MuSCs were labeled with GFP and tdT, which showed yellow fluorescence; while Gli1- MuSCs were only labeled with tdT (red). Next, we performed genetic lineage tracing experiments using the newly generated Gli1-CreERT2;

R26-eGFP; Pax7-DreERT2; R26-RSR-tdT mice by TAM induction. We observed the presence of GFP+tdT+ (Gli1+, Yellow) MuSCs in the muscle (**revised Supplementary Fig. 8e**). Then, we sorted GFP+tdT+ (Gli1+, Yellow) and GFP-tdT+ (Gli1-, Red) MuSCs and perform transplantation experiments, respectively (**revised Supplementary Fig. 8f**). Gli1-CreERT2; R26-DTA, NOD-SCID and DMD mice were used as recipient mice, respectively. Frozen sections of TA muscles being transplanted were subjected for fluorescence imaging analysis. TA transplanted with GFP+tdT+ (Gli1+, Yellow) MuSCs exhibited a higher number of fluorescently labeled muscle fibers, providing evidences of the stronger *in vivo* regenerative capacity as indicated by bigger size of the regenerated myofibers (**revised Figs. 5 and 6**). In the revised manuscript, we performed dual recombinases-mediated lineage tracing and clarified the fate of Gli1- and Gli1+ MuSCs, of which conclusions are mostly consistent with the ones reported in our original submission. Therefore, we also provide new figures (**revised Figs. 5 and 6** in the revision, in replacement of original **Fig. 5** in our original submission) for better demonstration of the data.

8. In Figure 6 panel L and M and the text line 343, the authors say that over 50% of MuSCs in the contra lateral leg of an injured mouse were Gli1+, in comparison to uninjured mice which is 13%. However, when looking at the data even a cursory glance shows that the number of Gli1+ cells in the CSC condition is nowhere near 50% despite the claims. In Figure 6 panel L, the total number of MuSCs/section is about 150 according to the graph, and the total number of Gli1+ cells in the CSC is generously 40 cells, again according to the graph, therefore the percentage of Gli1+ MuSCs is at most 26.67% ($40/150 * 100 = 26.67$) which is only half of the number the authors are reporting in the text and in Figure 6 panel M.

Response: We apologize for the confusion. What we meant to say was that the over 50% of the newly generated MuSCs induced by the injury at the contralateral leg was Gli1+ (**revised Fig. 7m, revised Supplementary Fig. 10j**). This is consistent with our notion that Gli1+ MuSCs are more active than Gli1- MuSCs, therefore, they are easier to enter the G_{Alert} state. We have changed the wording of the description of the results in the text and figure legend to further clarify our point.

9. It is not clear if the Gli1+ cells which are a very small percentage are contaminants or that their presence in scRNA-seq is due to contaminating ambient RNA. The authors could also run RNA velocity on the single cell data to see if their so called Gli1+ cells originate from Gli1- or vice versa.

Response: We thank the reviewer for the suggestion. We have run RNA velocity analysis. The RNA velocity analysis results suggested that the conversion potential between Gli1+ and Gli1- MuSCs is low (**Referee Fig. 2, revised Fig. 1i**).

Referee Fig. 2 RNA velocity analysis of single-cell RNA sequencing (scRNA-seq) data.

Minor concerns:

1. Could the authors clarify which Vcam1 antibody was used for which experiments

Response: We used PE/cy7 anti-mouse VCAM1 in the MitoTracker staining experiments, and APC anti-mouse VCAM1 in the other experiments. The above information has been included in the methods section in the revised manuscript.

2. The authors should include a control experiment where after FACS sorting tdT+ cells to stain them for Pax7, to confirm the percentage of sorted cells that are actually MuSCs

Response: We are grateful to the reviewer for the valuable suggestion. We performed Pax7 staining on all isolated MuSCs. Over 95% of the cells were Pax7+ (**revised Fig. 3b, c**), confirming that the isolated cells are indeed MuSCs.

3. Line 140-142 says "Immunofluorescence staining and FACS analysis on TA muscle sections and MuSCs showed that over 85% of tdT+ (Pax7+) cells were expressing Pax7...". The way this is phrased indicates that only 85% of the tdT+ cells are Pax7+, yet in the supplemental figure 2 the graph says that 85% of Pax7+ cells are tdT+, which one is correct?

Response: We apologize for the confusion. The correct description is: "85% of Pax7+ cells are tdT+." We have made the correction in the revised manuscript.

4. Figure 2 Panel E is not very representative of the data presented

Response: In the revised manuscript, we have included a more representative image in revised **Fig. 2e**.

5. Figure 2 Panel F, the overlay panel is brighter than the individual channels

Response: We have replaced the images in **revised Fig. 2g** to make all images at the same brightness.

6. To expand on the reported increase in differentiation, a western blot of differentiated tdT- and tdT+ myotubes for Gli1, MyHC, MyoD, and myogenin would be informative

Response: We have incorporated Western blot data of Gli1, MyHC, MyoD, and myogenin from differentiated tdT- and tdT+ myotubes. Upon differentiation, myotubes differentiated from tdT+ (Gli1+) MuSCs expressed higher level of MyHC and MyoG compared to myotubes differentiated from tdT- (Gli1-) MuSCs. Gli1 expression was not detected after differentiation (**Referee Figure 3, revised Fig. 3j**). These experimental results support that tdT+ MuSCs have superior differentiation potential.

Referee Fig. 3 Western blot for Gli1, MyHC, MyoD, MyoG and GAPDH from tdT+ (Gli1+) and tdT-(Gli1-) MuSCs induced myotubes.

7. Figure 5 Panel I, legend says n=6-10 mice used, however in the tdT- cells there are 10 data points in time to exhaustion and 11 data points in distance traveled. Further, the tdT+ cells have 11 data points.

Response: We apologies for this error. Indeed, we performed the physiological experiments using more mice. To make all the results consistent, we now include 10 data points for the figures in the revised manuscript.

8. Figure 5 Panel E, binning the CSA would be more informative

Response: As suggested by the reviewer, we binned the CSA data and added the CSA distribution graphs for Gli1-creERT2; R26-DTA uninjured and injured (14 days post-injury) samples (**revised Fig. 5d**).

9. Figure 5 Panel H, add the weight of the uninjured TA as an extra control

Response: As requested by the reviewer, we included the weight of uninjured TA as an extra control in **revised Fig. 6b**.

10. Figure 6 Panel B, the staining is very unconvincing. The Pax7 stain looks to be background

Response: Following the reviewer's suggestion, we re-stained the section and included the new imaged in **revised Fig. 7b**.

11. In the methods section, it is mentioned that isolated fibers were either fixed immediately after isolation or were cultured for 48 hours. However, there are no experiments performed on cultured myofibers

Response: We apologize for the missing data. We have now included the data from cultured myofibers in **revised Supplementary Fig. 7f-i**.

Reviewer #2 (Remarks to the Author):

Skeletal muscle is capable of robust regeneration thanks in part due to a rare population of skeletal muscle stem cells (MuSCs), also known as satellite cells for their satellite position to the myofibre, and underneath the basal lamina. Recent studies demonstrate substantial heterogeneity of MuSCs at various levels, including expression of transcription factors and surface markers, clonal ability, transplantation efficacy and functional responses to environmental stress. It is still unclear to what extent MuSCs are heterogeneous and what the functional implications of MuSC heterogeneity may be.

This is an interesting manuscript that has described a subset of MuSCs that uniquely express the GLI family transcription factor GLI1. Using genetic tools in the mouse, the authors show that approximately 15% of MuSCs express GLI1, depletion of the GLI1-expressing (GLI1(+)) MuSC population leads to delayed skeletal muscle regeneration, and that GLI1(+) MuSCs transplant more efficiently than GLI1(-) MuSCs. The authors further characterize GLI1(+) MuSCs as 'primed', characterized by increased mTORC1 signaling, increased cell size and mitochondrial numbers, reminiscent of a recently characterized 'Galert' phenotype of MuSCs that exist in skeletal muscle contralateral to acute injury. The authors conclude that GLI1(+) MuSCs exist as a sentinel population of MuSCs that rapidly respond to muscle injury.

There are many strengths to the manuscript as follows:

1. Elegant use of multiple genetic mouse models convincingly support the authors conclusions of a GLI1(+) MuSC population, and that this population of MuSCs contribute to MuSC regeneration.
2. To my knowledge the GLI1 member of the GLI family transcription factors has not been studied in MuSCs, although additional members (GLI3 and GLI2) have recently been shown to regulate quiescence and activation, respectively.
3. The notion of a 'sentinel' population of MuSCs (in the absence of acute damage elsewhere) is interesting and new.
4. In general, the authors conclusions are supported by high quality, convincing data throughout (but with some exceptions stated below).

We deeply appreciate the reviewer's thoughtful summarization of the strength of the manuscript and all the valuable suggestions to improve the manuscript.

There are however a number of important weaknesses that the authors need to address to support their conclusions:

1. Very recently, two papers have been published establishing cilia as important regulators of MuSC activity. Primary cilia function as hubs of hedgehog signaling, the effects of which are mediated by the GLI family of transcription factors. GLI3 is required for maintaining MuSC quiescence, and MuSCs lacking GLI3 enter the Galert state in the absence of injury via activation of mTORC1 signaling (PMID: 35803939). GLI2 is transiently upregulated upon MuSC activation, and aging MuSCs are defective for both primary cilium and GLI2 expression at basal state and after acute injury (PMID: 35301320). In light of these recent contributions, a question of novelty of the present work arises. In my opinion, the current manuscript has a strong foundation to significantly contribute new knowledge, but needs to address the following recommendations:

a. The authors have cited both of the papers I highlight above, but in my opinion these citations come too late in the manuscript. The Introduction of the manuscript should include a stronger introduction to the GLI family of TFs (page 4, last paragraph), and cite the current knowledge about cilia, HH signaling and GLI family of transcription factors in muscle stem cells and in myogenesis. An introduction providing the current knowledge will put the authors in a stronger position to highlight their new questions and novel contributions.

Response: We deeply appreciate the reviewer's great suggestions to improve the manuscript. We have polished the introduction of the manuscript as suggested by the reviewer.

b. To my knowledge, GLI1 is primarily a transcriptional activator, while GLI3 is a transcriptional repressor or activator. Since the absence of GLI3 was shown to cause MuSCs to adopt a Galert state (PMID: 35803939), I would like to know Gli3 expression levels in the subset of GLI1(+) MuSCs. If GLI1(+) MuSCs do express Gli3, how do the authors reconcile?

Response: We thank the suggestions of the reviewer. Gli3 is barely detected in Gli1+ MuSCs illustrated by bulk mRNA sequencing and RT-qPCR. In contrast, Gli3 expression level was significantly higher in Gli1- MuSCs (**revised Supplementary Fig. 6c, e**). The mutually exclusive expression pattern of Gli1 and Gli3 is consistent with the knowledge that Gli3 represses the expression of Gli1. Brun et al reported that knocking-out of Gli3

caused MuSCs to adopt a G_{alert} state (Brun et al, 2022, PMID: 35803939). Our results provide further evidences to suggest that the expression of Gli3 in Gli1+ MuSCs may repress the expression of Gli1, generating the Gli3 null MuSCs under physiological conditions. The mutual regulation of Gli1 and Gli3 in MuSCs is an interesting topic for further investigation.

2. I would like to see the GLI1(+) MuSCs characterized in greater detail with respect to their phenotype as follows:

a. Given the link to mTORC1 signaling and Galert, what are levels of protein synthesis in GLI1(+) vs GLI1(-) MuSCs, potentially addressed by puromycin incorporation assays.

Response: We deeply appreciate the reviewer's suggestion. We performed puromycin incorporation experiment and observed that Gli1+ MuSCs exhibit higher levels of protein synthesis compared with Gli1- MuSCs in both QSCs and ASCs (**revised Supplementary Fig. 9c, Referee Fig. 4**). This finding is consistent with the elevated mTOR signaling observed in Gli1+ MuSCs.

Referee Fig.4 a and b Western blot for Puromycin and GAPDH from tdT+(Gli1+) and tdT-(Gli1-) QSCs and ASCs.

b. What is the relative expression of Pax7/PAX7, CD34, Myf5/MYF5 and MyoD/MYOD in GLI1(+) vs GLI1(-) MuSCs, potentially addressed with the authors existing sequencing datasets for transcript levels and by western blotting or immunofluorescence with antibodies against PAX7, CD34, MYF5, MYOD.

Response: We thank the reviewer for this suggestion. Following the reviewer's suggestion, we performed RT-qPCR and checked the sequencing data for the expression level of Pax7, CD34, Myf5 and MyoD. The expression levels of Pax7, CD34, Myf5, and

MyoD in Gli1- and Gli1+ MuSCs were similar (**revised Supplementary Fig. 6d, f**). To further validate these results, we performed immunofluorescence staining, which also showed the similar expression levels of these genes in Gli1+ and Gli1- MuSCs (**revised Fig. 3b, revised Supplementary Fig. 6a**).

c. What is the relative ability of GLI1(+) MuSCs vs GLI1(-) MuSCs to contribute to self-renewal, a defining property of stem cells. This is potentially addressed *in vivo* by transplantation assays, followed by donor cell contribution to the PAX7(+) MuSC pool. In my opinion, this additional knowledge is essential to support the authors conclusions that GLI1(+) MuSCs have enhanced regeneration after acute injury and after engraftment.

Response: We thank the reviewer for pointing out this important issue. We agree with the reviewer that tracking both Gli1+ and Gli1- MuSCs after transplantation will greatly strengthen the notion that Gli1+ MuSCs support muscle generation better. In order to track both Gli1- and Gli1+ MuSCs, we generated a new mouse strain (Gli1-CreERT2; R26-eGFP; Pax7-DreERT2; R26-RSR-tdT) using a new genetic lineage tracing system that utilizes Dre and Cre recombinases to examine Gli1- and Gli1+ MuSCs. As shown in **revised Supplementary Fig. 8d**, Gli1+ MuSCs were labeled with GFP and tdT, which showed yellow fluorescence; while Gli1- MuSCs were only labeled with tdT (red). Next, we performed genetic lineage tracing experiments using the newly generated Gli1-CreERT2; R26-eGFP; Pax7-DreERT2; R26-RSR-tdT mice by TAM induction. We observed the presence of GFP+tdT+ (Gli1+, Yellow) MuSCs in the muscle (**revised Supplementary Fig. 8e**). Then, we sorted GFP+tdT+ (Gli1+, Yellow) and GFP-tdT+ (Gli1-, Red) MuSCs and perform transplantation experiments, respectively (**revised Supplementary Fig. 8f**). NOD-SCID mice was used as recipient mice. Frozen sections of TA muscles being transplanted were subjected for fluorescence imaging analysis. TA transplanted with GFP+tdT+ (Gli1+, Yellow) MuSCs exhibited a higher number of fluorescently labeled muscle fibers, providing evidences of the stronger *in vivo* regenerative capacity as indicated by bigger size of the regenerated myofibers (**revised Fig. 6**). Next, Pax7 staining was performed with muscle sections harvested 30 days after transplantation. Immunofluorescence staining for Pax7 showed that nuclear-localized Pax7 expression was observed in the tissue sections of the two groups of recipient mice (**Supplementary Fig. 8g**). However, we found that there was no significant difference in the number of Pax7-expressing cells, indicating that the two cell populations may have similar self-renewal ability. Although Gli1+ MuSCs are able to contribute more myofibers during

regeneration, which indicates stronger proliferation and differentiation abilities, there is no difference in self-renewal ability between Gli1+ and Gli1- MuSCs.

3. Related to Figure 3G, the authors conclude that GLI1(+) MuSCs have enhanced proliferation and enhanced differentiation. However, these two processes are mutually exclusive given that differentiating MuSCs exit the cell cycle to initiate fusion.

a. Given the experimental setup (2days proliferation medium, 2 days differentiation medium), it is likely that enhanced proliferation in the first two days gives rise to more myogenic progenitors that can give rise to a greater fusion index.

Response: We thank the reviewer for giving us the chance to explain the experiment in detail. We totally agree with the reviewer that the cell number at the initial stage of differentiation is critical. In fact, we performed the differentiation experiments using the same number of cells at the initiation stage of differentiation to avoid the potential complication from the difference of cell proliferation ability. We clarified this point in the revised manuscript in both the main text and the figure legend. We also performed Western blot to further confirm the differentiation ability of Gli1+ and Gli1- MuSCs. Higher MyHC and MyoG expression level and lower MyoD level were observed in differentiated myotubes from Gli1+ MuSCs (**revised Fig. 3j**), suggesting the enhanced differentiation ability of Gli1+ MuSCs.

b. If GLI1(+) MuSCs have enhanced differentiation, one might expect they precociously differentiate at the expense of proliferation. Precocious differentiation is potentially addressed using antibodies against Myogenin at early timepoints prior to fusion.

Response: We appreciate the valuable suggestions from the reviewer. We examined the expression of MyHC and Myogenin in QSCs, ASCs and Myotubes by RT-qPCR. Both Gli1+ and Gli1- MuSCs start to express MyHC and Myogenin in Myotubes (**Referee Fig. 5a, b**), suggesting that no obvious precocious differentiation in Gli1+ MuSCs. In addition, we performed immunofluorescence staining for MyoG in ASCs. The results demonstrated that Gli1+ MuSCs did not express MyoG before differentiation (**Referee Fig. 5b**). These results indicates that although Gli1+ MuSCs have a stronger ability to differentiate, they do not undergo premature differentiation before induction.

Referee Fig. 5 a and b *MyHC* and *MyoG* expression determined by RT-qPCR and normalized to *Gapdh* in QSCs, ASCs and 2 days-differentiated myotubes. **b** Immunofluorescence staining for *MyoG* (green), *tdT* (red) and DAPI (blue) in ASCs. Scale bars, 20 μ m.

c. If the authors want to demonstrate that GLI1(+) MuSCs have enhanced differentiation, the starting cell density should be the same at the step of changing to differentiation media.

Response: We indeed seeded the same number of MuSCs and start the differentiation. We have clarified this point in the revised manuscript.

4. A lot of data presented in Figure 5 are problematic for the following reasons:

a. The main problem being the Gli1(-) population cannot be tracked due to the lack of a fluorescent reporter. Although the data shows GLI1(+) cells contribute to regeneration after engraftment, strong conclusions about the efficiency of GLI1(+) vs GLI1(-) population cannot be made.

Response: We thank the reviewer for pointing out this important issue. We agree with the reviewer that tracking both Gli1+ and Gli1- MuSCs after transplantation will greatly strengthen the notion that Gli1+ MuSCs support muscle generation better. We have addressed this in question number 2c. Briefly, we generated a new mouse strain (Gli1-CreERT2; R26-eGFP; Pax7-DreERT2; R26-RSR-tdT) using a new genetic lineage tracing system that utilizes Dre and Cre recombinases to examine Gli1- and Gli1+ MuSCs (**revised Supplementary Fig. 8d-f**). Gli1-CreERT2; R26-DTA, NOD-SCID and DMD mice were used as recipient mice, respectively. Frozen sections of TA muscles being

transplanted were subjected for fluorescence imaging analysis. TA transplanted with GFP+tdT+ (Gli1+, Yellow) MuSCs exhibited a higher number of fluorescently labeled muscle fibers, providing evidences of the stronger *in vivo* regenerative capacity as indicated by bigger size of the regenerated myofibers (**revised Figs. 5 and 6**). In the revised manuscript, we performed dual recombinases-mediated lineage tracing and clarified the fate of Gli1- and Gli1+ MuSCs, of which conclusions are mostly consistent with the ones reported in the original submission. Therefore, we also provide new figures (**revised Figs. 5 and 6** in the revision, in replacement of **Fig. 5** in the original submission) for better demonstration of the data.

b. Figure 5d-e, multiple timepoints after engraftment would help. Additional markers for embryonic myosin heavy chain (embMHC; regenerating myofibres), PAX7 (donor MuSC population) would give more informative results.

Response: We appreciate the reviewer's valuable suggestions. We examined the regeneration process on day 5, 7, and 14 post injury. First, regeneration of TA muscle was evaluated 5, 7 and 14 dpi by Laminin staining. Mice transplanted with GFP+tdT+ (Gli1+) MuSCs displayed better regeneration, as indicated by a higher average area of myofibers than that of mice transplanted with GFP-tdT+ (Gli1-) MuSCs (**revised Fig. 5f**), suggesting that Gli1+ MuSCs have greater regenerative ability than Gli1- MuSCs. Next, given that a substantial generation of newly formed muscle fibers occurs around five days post-transplantation, we collected TA samples at this time point (5 and 7 dpi) and performed embMHC and Pax7 staining on frozen sections. The results demonstrated a higher number of embMHC+tdT+ muscle fibers and Pax7+tdT+ MuSCs in the TA transplanted with Gli1+ MuSCs (**revised Fig. 5g, h**), providing further evidence of the efficient promotion of skeletal muscle regeneration by Gli1+ MuSCs *in vivo*.

c. Essential to include non-engrafted control to observe the collapse of regeneration following DTA, and also to prove the DTA is working as expected to eliminate PAX7(+); GLI1(+) MuSCs.

Response: We thank the reviewer for the suggestion. We did not include the DTA alone control in the original manuscript due to the space limitation. Now we included this part in the revised manuscript. Impaired muscle regeneration was observed without transplantation, suggesting that the DTA mediated Pax7 elimination worked. These results are now included in **revised Fig. 5f** in the revised manuscript.

d. Notably the DTA experiments will eliminate this minor population of GLI1(+) MuSCs, but also essential supporting cells such as GLI1(+) FAPs, making interpretation of the results very difficult.

Response: We completely agree with the reviewer. The limitation of using Gli1-DTA mice is the elimination of all Gli1-expressing cells, including MuSCs, FAPs, and other cell types, which may all have some contributions to muscle regeneration. To address this problem, we utilized Gli1-CreERT2; R26-tdT mice and Gli1-CreERT2; R26-eGFP mice as donors. We sorted Gli1+ MuSCs from Gli1-CreERT2; R26-tdT mice and other Gli1+ cells (Gli1+ non-MuSCs) from Gli1-CreERT2; R26-eGFP mice. Both cell groups were transplanted to the mice induced DTA mediated Gli1+ cell elimination and the muscle regeneration was surveyed 14 days after transplantation. To roughly mimic the scenario *in vivo*, more other Gli1+ cells were used for transplantation. The ratio of Gli1+ MuSCs and Gli1+ other cells was 1:4 (**revised Fig. 5i**). Transplantation of Gli1+ MuSCs alone facilitates muscle regeneration as indicated by the formation of bigger tdT+ myofibers compared to no-transplantation control (**revised Fig. 5j, k**). In contrast, transplantation of Gli1+ other cells than MuSCs displayed modest improvement of muscle regeneration (**revised Fig. 5j, k**). TA transplanted with the group of Gli1+ MuSCs plus Gli1+ other cells showed robust muscle, with numerous tdT+ muscle fibers present, similar to the level of repair observed in the Gli1+ MuSCs alone group (**revised Fig. 5j, k**). These results further supported the crucial role of Gli1+ MuSCs in the process of muscle regeneration *in vivo*.

e. Figure 5c-e. I suppose the Gli1-CreERT2 will leave about 85% of the MuSC population intact, making interpretation of results difficult.

Response: We completely agree with the reviewer. After TAM induction, about 85% of the MuSCs population remains intact. They are predominant Gli1-. These cells participated in muscle regeneration. Muscle regeneration was impaired even though the Gli1- MuSCs remained intact. To exclude the possibility that other Gli+ cells than MuSCs make a significant contribution to muscle regeneration, we isolated and transplanted the Gli1+ non-MuSCs to the Gli1-CreERT2-DTA mice. Similar muscle regeneration defects were observed compared to no-transplantation control (**revised Fig. 5j, k**). Taken together, these results suggested that Gli1+ MuSCs are critical for proper muscle regeneration.

f. Essential to include non-engrafted control to observe the collapse of regeneration following irradiation, and also to prove the irradiation has eliminated the host MuSC population.

Response: We added the non-engrafted control as suggested by the reviewer. Severe muscle regeneration defects were observed after irradiation (**Referee Fig. 6**). Even at 30 days after injury, the regeneration was poor, suggesting the loss of MuSCs induced by irradiation.

Referee Fig. 6 Immunostaining for Laminin in TA tissue sections from Nod-Scid mice after CTX-induced injury. Scale bar, 50 μ m.

g. Figure 5g. I would normally insist to please report the number of donor derived myofibres as a percentage of total myofibres, but again the Gli1(-) population cannot be tracked. The efficiency of engraftment is important to understand Figure 5i.

Response: We thank the reviewer for pointing out this important issue. As explained above, we generated a new mouse line Gli1-CreERT2; R26-GFP; Pax7-DreERT2; R26-RSR-tdT to visualize both Gli1+ and Gli1- MuSCs simultaneously. Both Gli1+ and Gli1- MuSCs were isolated and transplanted to the recipient NOD-SCID or DMD mice, respectively. Gli1+ MuSC transplantation resulted in enhanced muscle regeneration compared with that from Gli1- MuSC transplantation (**revised Fig. 6**). The engraftment efficiency of Gli1+ and Gli1- MuSCs were analyzed. Laminin staining of TA muscle cryosections at 30 dpi in NOD-SCID or DMD mice showed that around 482 or 341 fibers were contributed by GFP+tdT+ (Gli1+) MuSCs during the regeneration process, whereas GFP-tdT+ (Gli1-) MuSCs contributed to approximately 224 or 108 fibers, respectively (**revised Fig. 6e, h**). Due to the higher engraftment efficiency of Gli1+ MuSC, the mice displayed improved muscle functions after transplantation.

h. The functional recovery reported in Figure 5i is remarkable given the experimental setup. It's essential to have appropriate controls, such as the PBS control in Figure 5i.

Response: We appreciate the suggestions from the reviewer and have incorporated the control group data with PBS injection into **revised Fig. 6c**.

5. Engraftment of donor cells into the Dmdmdx mouse model confers some advantages in that both GLI1(+) and GLI1(-) engraftment efficiency can be reported by numbers of donor derived myofibres expressing dystrophin. However, the dystrophin immunolabeling in Supplementary Figure 5h lower panels needs to be improved. The lower panels should look like the upper panels with respect to dystrophin, and the presented data raises questions whether the number of dystrophin (+) myofibres counted is accurate. The results should also show whether 18Gy irradiation eliminated endogenous MuSCs.

Response: We thank the reviewer for the valuable suggestions. As described above, we are able to visualize both Gli1+ and Gli1- MuSCs using the new mouse line Gli1-CreERT2; R26-GFP; Pax7-DreERT2; R26-RSR-tdT. Gli1+ (Yellow) and Gli1- (Red) MuSCs were transplanted to DMD recipient mice, respectively. We have repeated the dystrophin staining in **revised Fig. 6i, j**.

Due to the ongoing muscular fiber damage caused by the lack of dystrophin in DMD mice, exposing them to irradiation before transplantation would cause severe muscle injury that is challenging to repair. Therefore, in the repeated transplantation experiment, we opted not to irradiate DMD mice, but to directly transplant the donor cells. To verify the efficacy of irradiation in eliminating endogenous MuSCs, we subjected NOD-SCID mice to 18Gy X-ray irradiation. The regeneration status of non-transplantation control after irradiation was shown above in **Referee Fig. 6**. The muscle regeneration was severely impaired by irradiation, suggesting the elimination of endogenous MuSCs.

Minor points that don't require experimental revisions:

1. Pg 10 line 203 and Supplementary Figure 6 title. The data presented does not support the conclusion that GLI1(+) MuSCs are 'indispensible' for regeneration. The Figure 5 title is more appropriate heading given the results presented.

Response: We have changed the conclusion as "Gli1+ MuSCs are critical for proper muscle regeneration." in the revised manuscript.

2. Figure legends for Figure 5b-c are not correct, I think c refers to b and vice versa.

Response: We apologize for the error. We have made corrections and double checked that all the legends are right in the revised manuscript.

3. To increase readability, the authors could consider supplementing immunofluorescence images with short descriptors of experimental design (eg/ timepoints after ctx injury, age of mice, donor MuSC identity) underneath the images.

Response: We thank the reviewer for helping us improving the readability of the manuscript. We have added short descriptions of experimental design under each image.

4. Figure 3h, please explain the Fusion index in the y-axis (what are the results a percentage of?)

Response: We apologize for the confusion. The fusion index was calculated as the percentage of total nuclei incorporated into myotubes vs. the total number of nuclei. A higher fusion index indicates a higher degree of cell differentiation. Myotubes were identified by MyHC staining.

Reviewer #3 (Remarks to the Author):

Summary

In this manuscript, Peng et al. identified a novel subpopulation of muscle stem cells marked by the expression of Gli1. By scRNA-seq on skeletal muscle isolated from a conditional and inducible mouse model, in which all Gli1-expressing cells are labeled by the tdTomato (tdT) upon tamoxifen induction (Gli1+ tdT+), along with immunostaining on muscle sections and isolated myofibers, the authors show that around 10% of muscle stem cells (MuSCs) are expressing Gli1. Analyzing a cleverly designed mouse model that allows tracking Gli1 expression in muscle stem cells specifically (Pax7+ Gli1+ tdT+), as well as the Gli1-lacZ mouse line, further confirms their results. To characterize the Gli1+ cell population, the authors perform bulk RNA-seq analysis and in vitro assays on Gli1- and Gli1+ MuSCs and show that Gli1+ MuSCs exhibit enhanced proliferation and differentiation capacity. Then, they analyze the regenerative capacity of the Gli1+ MuSCs through a series of in vivo experiments. Cardiotoxin-induced muscle injury reveals that Gli1+ MuSCs participate to muscle regeneration and depleting this cell population alters the regeneration process. Transplantation experiments to show that Gli1+ MuSCs exhibit a higher ability to regenerate. Finally, the authors show that Gli1+ MuSCs mimic the GAlert state, conferring them a faster ability to activate and regenerate.

Critiques/Comments/Suggestions

1. The authors claim the existence of a subpopulation of Gli1-expressing muscle stem cells. However, the scRNA-seq data do not identify a distinct cluster for Gli1-expressing cells from FACS-isolated CD45- CD31- Sca1- VCAM+ cells (Fig. 1g-i). Instead, they show that some cells within the muscle stem cell population are expressing Gli1. Thus, the term of 'subpopulation' should be modified. To this reviewer's point of view, Gli1 maybe marks a state of quiescence but does not define a subpopulation of MuSCs. Also, from the Fig. 1g, would it be possible to get the proportion of muscle stem cells that are expressing Gli1? Does it correlate with the proportions of Gli1+ tdT+ MuSCs observed in Fig. 2?

Response: We thank the reviewer for the suggestion. We changed the word "subpopulation" to "population" in the revised manuscript. The percentage of Gli1+ MuSCs identified by single cell sequencing (**revised Fig. 1i**) is 6.47%. The percentage of Gli1+ tdT+ MuSCs shown in Fig. 2 is 13.8%. The percentage of Gli1+ MuSCs identified by single cell sequencing is lower than that identified by lineage tracing. It is important to note that lowly expressed transcript may appear sparsely expressed due

to sampling biases, also known as dropout effects, in single-cell RNA sequencing data. While it may appear that Gli1 is only expressed in a small number of cells, it is likely that because the transcripts are not very abundant, they are not detected for every cell in which they are truly expressed.

2. The manuscript lacks mechanistic studies that link Gli1 expression to the cellular phenotypes of the Gli1+ MuSCs.

a. Although the bulk RNA-seq analysis of the Gli1/tdT+ MuSCs shows the upregulation of downstream mediators of Hedgehog signaling that might regulate their cellular phenotypes (Fig. 3c, Gli1, Ptch1, Hhip), none of them are explored in the study. What is the expression of Gli1 in quiescent and activated muscle stem cells? In cultured myoblasts and myotubes? How does the Hedgehog signaling behave during MuSC progression in the myogenic lineage? Is Hedgehog signaling involved in the cellular phenotypes of Gli1+ MuSCs?

Response: We thank the reviewer for the suggestion. RT-qPCR was performed in both Gli1+ and Gli1- MuSCs. Consistent with the RNA-seq results, Gli1 was only expressed in Gli1+ MuSCs, not in the Gli1- MuSCs. Hhip was upregulated in Gli1+ MuSCs. However, there was no significant difference in the expression level of Ptch1 (**revised Supplementary Fig. 6c, e**). What is the role of Hh signaling in Gli1+ MuSCs is an interesting question to pursue. It is well-known that Gli1 is a transcription factor and surrogate of the Hh signaling pathway. As a result, we believe that there is good reason to doubt that Hh signaling pathway may play an important role in the regulation of Gli1+ MuSCs. Previous studies have demonstrated the pivotal role of the Hh signaling pathway in muscle regeneration, involving in both canonical and non-canonical mechanisms. The Hh pathway only displays low activity only under homeostatic conditions in adult muscle, whereas acute injuries robustly activate Hh signaling (Kopinke et al, 2017, PMID: 28709001). This increase in Hh activation during an acute injury indicates that the Hh pathway is required for muscle regeneration. A recent study found that FAPs, not MuSCs, are the key cellular responder of DHH and that Hh activation has no impact on *in vitro* myogenesis of either C2C12s or primary MuSCs (Norris et al, 2023, PMID: 37355632). Recent study found that Gli3 is required for maintaining MuSCs quiescence, and MuSCs lacking Gli3 enter the G_{Alert} state in the absence of injury via activation of mTORC1 signaling (Brun et al, 2022, PMID: 35803939). In our experiments, we observed that Gli1 is predominantly expressed in freshly isolated Gli1+ MuSCs. Furthermore, Gli1 expression was downregulated in activated MuSCs and barely detectable in myotubes (**Referee Fig. 7**). The expression of Gli3 was barely detected in Gli1+ MuSCs, while its expression in Gli1- MuSCs was relatively high. The absence of Gli3 in QSCs induces G_{Alert} , in myogenic progenitors increase their proliferation and form larger myotubes

upon differentiation *in vitro*. All these results seem to suggest that Gli1+ MuSCs exhibit function and physiological properties, which is related to Gli3 low expression in uninjured muscle. Gli3 has pleiotropic roles during muscle stem cell progression through the myogenic lineage: controlling satellite cell quiescence and activation independently of Hh ligand receptor binding, while repressing Hh signaling target genes in myogenic progenitors to regulate their proliferation and differentiation (Brun et al, 2022, PMID: 35803939). Therefore, we think that Gli1+ MuSCs has pleiotropic roles during muscle stem cell progression through the myogenic lineage.

Referee Fig. 7 Relative mRNA levels of Gli1 and Gli3 in QSCs, ASCs and Myotubes.

b. Besides cell proliferation and differentiation, the transcriptome analysis on freshly FACS-isolated Gli1/tdT+ and Gli1/tdT- muscle stem cells also reveals gene ontology terms related to regulation of IGF transport and mTOR signaling (Fig. 2c, d). How do the authors link Gli1 to these signaling pathways? Does it stand true for the cultured MuSCs (myoblasts) and differentiated cells (myotubes)? Are the down-/up-regulated genes depicted in Fig. 2e still differentially expressed in cultured muscle cells? Especially, are there any differences in the Insulin-like growth factor and mTOR signaling activation? For instance, mTOR signaling is required for myogenic progenitor proliferation and fusion (10.1242/dev.172460; 10.1016/j.bbrc.2015.05.032). Increased mTOR activity in Gli1+ muscle cells would not only explain the GAlert phenotype in MuSCs but also the increased ability of the Gli1+ myogenic progenitors to proliferate and differentiate. The authors should assess the mTOR signaling activity in Gli1- and Gli1+ myogenic cell culture.

Response: We thank the reviewer for the valuable suggestion. GO analysis of transcriptomic data from Gli1+ and Gli1- muscle stem cells demonstrated that both mTOR and IGF signaling pathways were significantly upregulated in Gli1+ MuSCs. The genes shown significant expression changes in mTOR and IGF signaling pathways. Previous studies have revealed the existence of a Gli-mediated mTORC1 activation (Agarwal et al, 2013, PMID: 23580656; Zeng et al., 2017, PMID: 29065414; Klein et al., 2019, PMID: 31639285), in which Gli activates mTORC1 signaling by

down-regulating negative or up-regulating positive mTORC1 mediators. Hh signaling activates transcription of multiple members of the IGF pathway, most notably IGF-2, resulting in activation of the mTORC2-Akt signaling axis (Shi et al, 2015, PMID:25825734). IGF signaling has been shown that enhanced protein synthesis in muscle is mediated by activation of the PI3K-Akt-mTOR signaling pathway (Vyas et al., 2002, PMID:12107064). These works suggest Gli1 to be the potential activator of mTOR and IGF signaling. To further explain this, RT-qPCR was performed to detect the expression level of mTOR and IGF signaling-related genes in tdT⁺ and tdT⁻ MuSCs after culture. The results showed that the genes upregulated in freshly isolated stem cells exhibited a similar expression trend in cultured cells (**Referee Fig. 8a**). We also performed puromycin incorporation experiment and observed that Gli1⁺ MuSCs exhibit higher levels of protein synthesis compared with Gli1⁻ MuSCs in cultured cells (**revised Supplementary Fig. 9c, Referee Fig. 8b**). In addition, we performed pS6 staining on cultured myofibers, and quantified the ratio of pS6 in tdT⁺ and tdT⁻ MuSCs (**revised Supplementary Figs. 9b, 10h**). The results indicated that compared to Gli1⁻ MuSCs, Gli1⁺ MuSCs exhibited higher the ratio of pS6. Taken together, these findings are consistent with the elevated mTOR signaling observed in Gli1⁺ MuSCs.

Referee Fig. 8 a Relative mRNA levels of mTOR and IGF signaling-related genes in tdT⁺(Gli1⁺) and tdT⁻(Gli1⁻) MuSCs after culture. **b** Western blot for Puromycin and GAPDH from tdT⁺(Gli1⁺) and tdT⁻(Gli1⁻) MuSCs.

3. As mentioned in the introduction, the Hedgehog signaling effector Gli1 is expressed in stromal cells of many organs but it remains unclear whether Gli1 expression marks a subset of stem cells or progenitors. Using the tdT reporter to trace the Gli1⁺ muscle stem cell population, the authors show that Gli1⁺ MuSCs participate to muscle regeneration as around 75% of the newly formed myofibers are labeled by the tdT (Fig. 4). However, they do not assess whether the Gli1⁺ tdT⁺ cells can self-renew.

a. As they are more prone to proliferate, differentiate and fuse (Fig. 3), one could think

that this cell population resemble more a committed myogenic progenitor population that will be dedicated for rapid regeneration upon an injury, rather than a muscle stem cell population that will self-renew and repopulate the niche following the injury. To tackle this question, the authors should quantify the number of tdT- and tdT+ PAX7-expressing cells on cross-sections of regenerated muscles (Fig. 4).

Response: We thank the reviewer for the valuable suggestion. We quantified the number of tdT- Pax7+ and tdT+ Pax7+ cells in muscle section from Gli-CreERT2; Pax7 DreERT2; Ai66 mice 14 days after injury as suggested by the reviewer. The results indicated that the number of Gli1+ MuSCs accounted for approximately 17.4% of the total MuSCs, which was higher than the pre-injury proportion of approximately 13.8%, demonstrating the self-renewal ability of Gli1+ MuSCs. These results have been included in the revised manuscript as **revised Fig. 4g**.

b. PAX7-expressing cells should be also quantified on cross-sections of recipient muscle following the tdT- and tdT+ MuSC transplantation (Fig. 5). That way, the authors will determine whether the tdT+ cells represent a population of stem cells or a population of committed progenitor cells.

Response: We thank the reviewer for pointing out this important issue. In order to track both Gli1- and Gli1+ MuSCs, we generated a new mouse strain (Gli1-CreERT2; R26-eGFP; Pax7-DreERT2; R26-RSR-tdT) using a new genetic lineage tracing system that utilizes Dre and Cre recombinases to examine Gli1- and Gli1+ MuSCs. As shown in **revised Supplementary Fig. 8d**, Gli1+ MuSCs were labeled with GFP and tdT (red), which showed yellow fluorescence; while Gli1- MuSCs were only labeled with tdT. Next, we performed genetic lineage tracing experiments using the newly generated Gli1-CreERT2; R26-eGFP; Pax7-DreERT2; R26-RSR-tdT mice by TAM induction. We observed the presence of GFP+tdT+ (Gli1+, Yellow) MuSCs in the muscle (**revised Supplementary Fig. 8e**). Then, we sorted GFP+tdT+ (Gli1+, Yellow) and GFP-tdT+ (Gli1-, Red) MuSCs and perform transplantation experiments, respectively (**revised Supplementary Fig. 8f**). NOD-SCID mice was used as recipient mice. The number of tdT+ (Gli1+) Pax7+ and tdT- (Gli1-) Pax7+ MuSCs was quantified 30 days after transplantation as suggested by the reviewer. Immunofluorescence staining for Pax7 showed that nuclear-localized Pax7 expression was observed in the tissue sections of the two groups of recipient mice (**Supplementary Fig. 8g**). However, we found that there was no significant difference in the number of Pax7-expressing cells, indicating that the two cell populations may have similar self-renewal ability. Although Gli1+ MuSCs are able to contribute more myofibers during regeneration,

which indicates stronger proliferation and differentiation abilities, there is no difference in self-renewal ability between Gli1⁺ and Gli1⁻ MuSCs.

c. Although they perform flow cytometry analysis to show that there is still a proportion of CD45⁻ CD31⁻ Sca1⁻ VCAM⁺ mononucleated cells that express tdT, they cannot conclude if this population maintains PAX7 expression (VCAM marks both the PAX7⁺ muscle stem cells and the differentiated Myogenin⁺ muscle cells). Again, quantifying the number of tdT⁻ and tdT⁺ PAX7-expressing cells would be much better to characterize each cell type.

Response: We thank the reviewer for the suggestion. Immunofluorescence staining was performed for Pax7 in the sorted tdT⁺ and tdT⁻ MuSCs. Pax7⁺ cells accounted for over 95% of cells in each cell population (**revised Fig. 3b, c**). Furthermore, RT-qPCR was performed to detect the expression level of Myogenin in tdT⁺ and tdT⁻ cells after sorting. Myogenin was barely detected in these cells (**revised Supplementary Fig. 6f**), further support that the tdT⁺ and tdT⁻ cells obtained by FACS sorting are muscle stem cells.

4. Regarding the transplant experiments (Fig. 5), a major concern is that there is no means to track the tdT⁻ muscle stem cell population in order to compare their engraftment capability with the tdT⁺ MuSCs. Therefore, it is difficult to conclude anything on the comparison made.

a. This reviewer would suggest either to use a double-fluorescent cre-reporter mouse (such as the mT/mG) instead of the tdT reporter mouse or to cross their mice (Gli1-CreERT2; Pax7-DreERT2; Ai66) with the CAG-GFP transgenic mice. Thus, the authors will track both the Gli1⁻ (mT⁺ or GFP⁺) and Gli1⁺ (mG⁺ or GFP⁺ tdT⁺) cells following their transplantation.

Response: We really appreciate the suggestion made by the reviewer. We totally agree with the reviewer that tracking Gli1⁻ MuSCs after transplantation is critical to sort out the functions of Gli1⁺ MuSCs. As mentioned above, in order to track both Gli1⁻ and Gli1⁺ MuSCs, we generated a new mouse strain (Gli1-CreERT2; R26-eGFP; Pax7-DreERT2; R26-RSR-tdT) using a new genetic lineage tracing system that utilizes Dre and Cre recombinases to examine Gli1⁻ and Gli1⁺ MuSCs. As shown in **revised Supplementary Fig. 8d**, Gli1⁺ MuSCs were labeled with GFP and tdT, which showed yellow fluorescence; while Gli1⁻ MuSCs were only labeled with tdT (red). Next, we performed genetic lineage tracing experiments using the newly generated Gli1-CreERT2; R26-eGFP; Pax7-DreERT2; R26-RSR-tdT mice by TAM induction. We observed the presence of GFP⁺tdT⁺ (Gli1⁺, Yellow) MuSCs in the muscle (**revised Supplementary Fig. 8e**). Then, we sorted GFP⁺tdT⁺ (Gli1⁺, Yellow) and GFP⁻tdT⁺

(Gli1-, Red) MuSCs and perform transplantation experiments, respectively (**revised Supplementary Fig. 8f**). Gli1-CreERT2; R26-DTA, NOD-SCID and DMD mice were used as recipient mice, respectively. Frozen sections of TA muscles being transplanted were subjected for fluorescence imaging analysis. TA transplanted with GFP+tdT+ (Gli1+, Yellow) MuSCs exhibited a higher number of fluorescently labeled muscle fibers, providing evidences of the stronger *in vivo* regenerative capacity as indicated by bigger size of the regenerated myofibers (**revised Figs. 5 and 6**). In the revised manuscript, we performed dual recombinases-mediated lineage tracing and clarified the fate of Gli1- and Gli1+ MuSCs, of which conclusions are mostly consistent with the ones reported in the original submission. Therefore, we also provide new figures (**revised Figs. 5 and 6** in the revision, in replacement of **Fig. 5** in the original submission) for better demonstration of the data.

b. In Fig. 5g, the muscle picture of the NOD-SCID recipient mouse is not injured (no centrally located nuclei, no sign of regeneration) while it should be. Yet tdT+ myofibers are observed. This observation supports the idea that Gli1+ tdT+ cells are differentiating faster compared to the tdT- and are pro-fusogenic. Please revise your figure. As well and for each engraftment experiment, please provide a representative picture of the full muscle section.

Response: We thank the reviewer for this suggestion. The muscle tissues were harvested 30 days after injury. Some regions of the injury site have reached a near complete regeneration and centrally located nuclei were not observed. We re-analyzed the whole set of sections from the transplantation experiments and found that there were regions containing centrally located nuclei. As suggested by the reviewer, tdT+ (Gli1+) MuSCs transplantation indeed resulted in more complete regeneration as indicated by the reduced number of myofibers containing the centrally located nuclei (**revised Fig. 6d**). These results suggest that tdT+ (Gli1+) MuSCs have better ability to support regeneration. As suggested by the reviewer, we now include the representative picture of the full muscle section for each transplantation experiment in the revised manuscript as **revised Fig. 6d**

c. The authors transplanted the tdT- and tdT+ MuSCs within irradiated and injured muscle of mdx mice. That way, they can track the fate of the engrafted MuSCs without relying on a reporter (Supp Fig. 6). These panels should probably appear in the main figures as it supports their conclusion. However, as mentioned above, quantifying the number of tdT- and tdT+ PAX7-expressing cells is required to conclude on the status of each cell type. As well, can the author add a Laminin immunostaining to show the surrounding mdx myofibers? Also, please provide a better staining for dystrophin in the

tdT+ MuSC condition.

Response: We thank the reviewer for this suggestion. In the revised manuscript, Immunofluorescence staining was performed for Pax7 in the sorted tdT+ and tdT- MuSCs. Pax7+ cells accounted for over 95% of cells in each cell population (**revised Fig. 3b, c**). Furthermore, we performed Laminin immunostaining to visualize the myofibers (**revised Fig. 6g, h**). We have also repeated the dystrophin staining for **revised Fig. 6i, j**.

5. The authors only quantify the proportions of the tdT- and tdT+ cell populations in Supp. Fig. 4d, Fig. 6h and Supp. Fig. 7c. Thus, it looks like the tdT+ cells are increasing during the first few days of regeneration while the tdT- are decreasing. One could expect that both cell types are increasing in numbers but the tdT+ proliferate faster than the tdT-. Providing the absolute numbers for each cell types would clarify their results and conclusion.

Response: We thank the reviewer for this suggestion. To determine the absolute numbers of Gli1- and Gli1+ MuSCs at various time points (0, 3, 5, 7, and 14 days post injury), we utilized a magnetic bead-based quantification method, which is commonly used in absolute quantification of immune cells (Duggan et al, 2017, PMID: 28087538; Ho et al, 2018, PMID: 29844427). A predetermined quantity of magnetic beads was mixed with digested single-cell suspension. After flow cytometry antibody staining, FACS sorting was performed and the absolute numbers of Gli1+ and Gli1- MuSCs were calculated based on the magnetic bead readings and cell counts. The results demonstrated that Gli1+ MuSCs experienced a significant increase in cell number during the early stage of regeneration. Although the number of Gli1+ MuSCs decreased in the later stages of injury, it was still higher than the D0 baseline. As suggested by the reviewer, though the percentage of Gli1- MuSC decreased at the early stage of the muscle regeneration, the absolute number of the cells increased. We now added these results in the revised manuscript as **revised Supplementary Table 1**

6. Figure 2f: Would it be possible to get quantify the number of PAX7+ tdT- and PAX7+ tdT+ muscle stem cells per myofiber?

Response: We thank the reviewer for the suggestion. We isolated individual muscle fibers and conducted a thorough scanning to quantify the number of Pax7+ tdT+ and Pax7+ tdT- cells (**revised Supplementary Fig. 3b, c**), consistent with our other results, the tdT+ (Gli1+) MuSCs accounted for about 12% of the MuSCs per fiber.

7. Lines 206-212: This paragraph should appear earlier in the 'results' section. Given

that CreERT2 insertion in the Gli1 allele can affect Gli1 expression, it seems important to mention that the Gli1-CreERT2 allele is kept heterozygous throughout the study, especially for the in vivo Gli1+ tdT+ cell tracing and regeneration experiments.

Response: We thank the reviewer for the valuable comments. In the revised manuscript, we have moved this paragraph to the Results section and emphasized throughout the study that we utilized heterozygous Gli1-CreERT2 mice.

8. Lines 336-338: “tdT+ (Gli1+) MuSCs display rapid cell-cycle entry, increased proliferation and augmented self-renewal and markedly enhanced regenerative capacity in the contralateral TA muscle (Supplementary Fig. 7e-i)”. This is a very strong statement. The authors did not assess the self-renewal ability of the tdT+ cells. Moreover, they cannot assess their regenerative capacity as the experiment was performed at day 1 post-injury. Please modify/clarify.

Response: We thank the reviewer for the suggestion. We have changed the description as “tdT+ (Gli1+) MuSCs display rapid cell-cycle entry, increased proliferation and elevated mTOR signaling in the contralateral TA muscle” in the revised manuscript.

9. a. Figure 6b: There is no PAX7 staining on the representative image selected. Given that the tdT labels all the Gli1-expressing cells independently of Pax7 expression, the cell shown in the image might be a PAX7- cell, such as a FAP that can also transit from G0 to GAlert (doi: 10.1038/nature13255). To avoid any confusion, the PAX7 staining should be redone/improved. As well, the authors could examine the expression of phospho-S6 in satellite cells from freshly isolated myofibers and quantify the number of pS6+ tdT+ and pS6+ tdT- MuSCs.

Response: We thank the reviewer for this suggestion. We replaced the Pax7 staining with a better image in the revised manuscript (**revised Fig. 7b**). We performed pS6 staining on freshly isolated and cultured myofibers, and quantified the ratio of pS6 in tdT+ and tdT- MuSCs (**revised Supplementary Figs. 9b, 10h**).

b. Figure 6e: There is only one replicate of each genotype shown and therefore, no quantification of the mitochondrial mass for each genotype. Quantifying the mean fluorescence intensity for 3 biological replicates would strengthen the data.

Response: We thank the reviewer for the suggestion. As shown in **revised Fig. 7e**, we have added more biological repeats and conducted Mean Fluorescence Intensity (MFI) analysis.

c. Figure 6i: If Gli1 expression is not maintained following MuSC commitment into the myogenic lineage, it is not surprising that the Gli1- myoblasts do not turn on the tdT

upon 4-hydroxytamoxifen treatment *in vitro*. Evaluating Gli1 expression seems to be important for the study (see comment 2a).

Response: We performed RT-qPCR and Gli1 staining of quiescent, activated MuSCs, and differentiated myotubes to investigate the expression pattern of Gli1. The results confirmed that Gli1 is expressed in quiescent MuSCs but is less expressed in activated and differentiated myotubes (**Referee Fig. 7**). To further address the potential conversion between Gli1⁺ and Gli1⁻ MuSCs, we performed transplantation experiments on Nod-Scid mice. TAM was injected intraperitoneally on the second day after tdT⁻ (Gli1⁻) MuSCs transplanting (**revised Supplementary Fig. 10a**). After TAM treatment, we did not observe tdT⁺ fibers or cells in the recipient mice, indicating that tdT⁻ (Gli1⁻) MuSCs do not convert to tdT⁺ (Gli1⁺) MuSCs *in vivo* (**revised Supplementary Fig. 10b**). Next, Gli1⁺ MuSCs were transplanted into the pre-irradiated TA muscles of Nod-Scid mice 1 day after CTX-induced injury (**Referee Fig. 9a**). TA muscles were harvested 30 days after transplantation, and the expression of Gli1 was examined by immunofluorescence staining. 30 days after the transplantation, as expected, many myofibers were also tdT⁺ (**Referee Fig. 9b**), suggesting that the transplanted tdT⁺ (Gli1⁺) MuSCs participate the muscle regeneration. We also observed Gli1 expression in almost all of tdT⁺Pax7⁺ MuSCs, suggesting that some of the transplanted MuSCs retain Gli1 expression. In addition to the Gli1⁺tdT⁺ MuSCs, population of tdT⁺ cells or tdT⁺ myofibers emerges that no longer expresses endogenous Gli1 protein. These results confirmed that tdT⁺ (Gli1⁺) MuSCs may convert to tdT⁻ (Gli1⁻) cells *in vivo*. We also performed RNA velocity analysis with our

single cells sequencing data. Collectively, these results suggested that the potential of conversion between Gli1+ and Gli1- MuSCs is low.

Referee Fig. 9 **a** Scheme of the experimental strategy. **b** Particular numbers (5,000 cells) of freshly isolated tdT+ (Gli1+) MuSCs from Gli1-CreERT2; R26-tdT mice were transplanted into pre-injured TA muscle of the NOD-SCID mice. TA muscles were harvested 30 days after transplantation. Scale bars, 50 μ m.

d. Figure 6l: It would be nice to see representative images for all the quantifications shown. That way, the data could be more appreciated.

Response: We thank the reviewer for this suggestion. We have now added representative images for all the quantification in **revised Supplementary Fig. 10i**.

e. Figure 6m: Legends are unclear. Shouldn't it be written uninjured and contralateral side instead of MuSCs and increased MuSCs?

Response: We deeply appreciate the reviewer's suggestion. What we meant to say was that the over 50% of the newly increased MuSCs induced by the injury at the contralateral leg was Gli1+ (**revised Fig. 7m**). This is consistent with our notion that Gli1+ MuSCs are more active than Gli1- MuSC, therefore, they are easier to enter the G_{Alert} state. We have changed the wording of the description of the results in the text and figure legend to further clarify our point. We also have changed the description in the legends as "uninjured and contralateral side" in the revised manuscript (**revised Supplementary Fig. 10j**).

REVIEWERS' COMMENTS

Reviewer #1 (Remarks to the Author):

The authors have reasonably addressed all my previous concerns.

Reviewer #2 (Remarks to the Author):

This is an interesting manuscript that characterizes a subset of MuSCs expressing GLI1. The original manuscript demonstrated that depletion of GLI1(+) MuSCs delays muscle regeneration, while GLI1(+) MuSCs transplant more effectively than GLI1(-) ones. The authors showed that these GLI1(+) MuSCs exhibit a 'primed' state, characterized by increased mTORC1 signaling, larger size, and more mitochondria. Overall, the authors proposed that GLI1(+) MuSCs serve as a rapid-response sentinel population to muscle injury.

The resubmitted manuscript is a significant improvement upon the original submission. The authors have thoughtfully addressed peer review, and have provided a substantial amount of new data to support their conclusions. The characterization of GLI1 expression at the protein level in MuSCs has been improved. The authors have used new genetic tools to trace the lineage of both GLI1(-) and GLI1(+) MuSCs, and have increased the use of engraftment assays to better define the activity of GLI1 MuSCs in a regenerative response (as opposed to other GLI1 expressing cells and GLI1(-) MuSCs). In my opinion, the authors have made a significant effort to address all of my concerns with new experiments, and have been forthcoming when results did not necessarily support lines of inquiry (eg/ GLI1-positive cells do not appear to have differential myogenic gene expression, nor have any difference in self-renewal capacity).

Given the overall contribution, I agree that an in depth analysis investigating the function of the Gli1 gene in MuSCs is better addressed by a separate study.

Altogether, the authors have more than satisfactorily addressed my concerns.

Minor concerns:

(1) the authors should describe the generation of Pax7DreERT2/+ mice in the materials and methods. I did find a schematic and initial characterization of the new genetic tool in supplementary information.

(2) I still have an issue with line 285 for the reason that all GLI1(+) cells, not just MuSCs, are ablated. This point is definitely made clear in the subsequent writing and experimentation, but should be clarified from the beginning.

Reviewer #3 (Remarks to the Author):

I commend the authors for providing a markedly improved manuscript. Congratulations on generating the Gli1-CreERT2; R26-eGFP; Pax7-DreERT2; R26-RSR-tdT mouse model to highlight the behavior of the Gli1+ 'sentinel' MuSC population post-engraftment! The authors have addressed pretty much all of my questions. I still have a few suggestions/comments.

In Supp Fig 6c, d, would it be possible to plot the data in a heatmap rather than bar graphs? Like this, all replicates could be shown.

In Fig 3h, please start the y-axis at 0 rather than 30. It will reflect better the data.

Could the author provide the rationale for using a ratio of 1:4 for the [Gli1+ MuSCs:Gli1+ other cells] transplantation? Regarding this experiment, how the GFP+ Gli1+ other cells were sorted? If the cells were sorted based on the GFP only, one could expect to have some MuSCs within this population (as 5% of Gli1+ cells express Pax7, Fig 1d-f). Did the authors observe any GFP+ myofibers?

Fig 4g, please show an example for each type of MuSCs (tdT- and tdT+).

In Fig 5f, please provide both the distribution and the average of myofiber CSA. Is it significantly different? What is the unit for the x-axis?

In Supp Fig 7i (figure and legend), do the data represent MuSCs per clone or per myofiber?

Although Gli1 transcript expression drops as MuSCs exit quiescence and enter the myogenic lineage, GLI1 protein expression seems to be maintained in MuSCs on myofibers cultured for 72h. Can the authors discuss on this?

Lines 313: equal volume PBS were transplanted into INJURED TA muscle [...] instead of "equal volume PBS were transplanted into TA muscle [...]"?

REVIEWERS' COMMENTS

Reviewer #1 (Remarks to the Author):

The authors have reasonably addressed all my previous concerns.

We would like to thank the reviewer for valuable comments and suggestions during the previous revision of the manuscript. These inputs have significantly contributed to enhancing the robustness of our reported findings.

Reviewer #2 (Remarks to the Author):

This is an interesting manuscript that characterizes a subset of MuSCs expressing GLI1. The original manuscript demonstrated that depletion of GLI1(+) MuSCs delays muscle regeneration, while GLI1(+) MuSCs transplant more effectively than GLI1(-) ones. The authors showed that these GLI1(+) MuSCs exhibit a 'primed' state, characterized by increased mTORC1 signaling, larger size, and more mitochondria. Overall, the authors proposed that GLI1(+) MuSCs serve as a rapid-response sentinel population to muscle injury.

The resubmitted manuscript is a significant improvement upon the original submission. The authors have thoughtfully addressed peer review, and have provided a substantial amount of new data to support their conclusions. The characterization of GLI1 expression at the protein level in MuSCs has been improved. The authors have used new genetic tools to trace the lineage of both GLI1(-) and GLI1(+) MuSCs, and have increased the use of engraftment assays to better define the activity of GLI1 MuSCs in a regenerative response (as opposed to other GLI1 expressing cells and GLI1(-) MuSCs). In my opinion, the authors have made a significant effort to address all of my concerns with new experiments, and have been forthcoming when results did not necessarily support lines of inquiry (eg/ GLI1-positive cells do not appear to have differential myogenic gene expression, nor have any difference in self-renewal capacity).

Given the overall contribution, I agree that an in depth analysis investigating the function of the Gli1 gene in MuSCs is better addressed by a separate study.

Altogether, the authors have more than satisfactorily addressed my concerns.

We greatly appreciate the reviewer's acknowledgement of the significance of the present study and the professional suggestions.

Minor concerns:

(1) the authors should describe the generation of Pax7DreERT2/+ mice in the materials and methods. I did find a schematic and initial characterization of the new genetic tool in supplementary information.

Response: We thank the reviewer for the suggestion. We have now described the generation of Pax7DreERT2/+ mice in the materials and methods in the revised manuscript (page 28, lines 560-564).

(2) I still have an issue with line 285 for the reason that all Gli1(+) cells, not just MuSCs, are ablated. This point is definitely made clear in the subsequent writing and experimentation, but should be clarified from the beginning.

Response: We agree with the reviewer. We have clarified that all Gli1(+) cells, not just MuSCs, are ablated in Gli1-creER; DTA mice from the beginning in the revised manuscript (page 14, lines 286-287).

Reviewer #3 (Remarks to the Author):

I commend the authors for providing a markedly improved manuscript. Congratulations on generating the Gli1-CreERT2; R26-eGFP; Pax7-DreERT2; R26-RSR-tdT mouse model to highlight the behavior of the Gli1+ 'sentinel' MuSC population post-engraftment! The authors have addressed pretty much all of my questions. I still have a few suggestions/comments.

We would like to thank the reviewer for the valuable comments and suggestions to improve our study. We respond to the specific comments below.

In Supp Fig 6c, d, would it be possible to plot the data in a heatmap rather than bar graphs? Like this, all replicates could be shown.

Response: We thank the reviewer for the suggestion. We have changed the style of Supp Fig 6c, d into a heatmap as you suggested. Please see the new submitted Supp Fig 6c.

In Fig 3h, please start the y-axis at 0 rather than 30. It will reflect better the data.

Response: We thank the reviewer for the suggestion. In the revised Fig 3h, we have adjusted the y-axis to begin at 0.

Could the author provide the rationale for using a ratio of 1:4 for the [Gli1+ MuSCs:Gli1+ other cells] transplantation? Regarding this experiment, how the GFP+ Gli1+ other cells were sorted? If the cells were sorted based on the GFP only, one could expect to have some MuSCs within this population (as 5% of Gli1+ cells express Pax7, Fig 1d-f). Did the authors observe any GFP+ myofibers?

Response: To effectively address this question and respond to the reviewer's inquiry, we will provide answers from the following perspectives:

- 1) To establish the transplantation ratio between Gli1+ MuSCs and Gli1+ other cells, we isolated hindlimb skeletal muscles from Gli1-CreERT2; R26-eGFP; Pax7-DreERT2; R26-RSR-tdT mice. We determined the ratio of Gli1+ MuSCs (GFP+tdT+) to Gli1+ other cells (GFP+tdT-) using cell sorting. The results showed a Gli1+ MuSCs: Gli1+ other cells ratio of 1:4. Consequently, we employed a 1:4 ratio for the transplantation experiments.
- 2) We sorted Gli1+ MuSCs (CD45-CD31-Sca1-VCAM1+tdT+) from Gli1-CreERT2; R26-tdT mice and Gli1+ other cells (Gli1+ non-MuSCs, CD45-CD31-GFP+tdT-) from Gli1-CreERT2; R26-eGFP; Pax7-DreERT2; R26-RSR-tdT mice.
- 3) TA transplanted with Gli1+ non-MuSCs did not exhibit any GFP+ myofibers.

Fig 4g, please show an example for each type of MuSCs (tdT- and tdT+).

Response: We thank the reviewer for the suggestion. We have now included each type of MuSCs (tdT- and tdT+) in revised Fig 4g.

In Fig 5f, please provide both the distribution and the average of myofiber CSA. Is it significantly different? What is the unit for the x-axis?

Response: As requested by the reviewer, we included both the distribution and the average of myofiber CSA in revised Supp Fig 8g, and it exhibits a significant difference between them. The unit for the x-axis is μm^2

In Supp Fig 7i (figure and legend), do the data represent MuSCs per clone or per myofiber?

Response: We corrected the mistake in the revised manuscript (page 13, lines 249). The data represented MuSCs per clone.

Although Gli1 transcript expression drops as MuSCs exit quiescence and enter the myogenic lineage, GLI1 protein expression seems to be maintained in MuSCs on myofibers cultured for 72h. Can the authors discuss on this?

Response: Thanks for your helpful comments. The Reviewer raised an issue regarding the “mismatch” between Gli1 mRNA level and protein levels. It is well known that the decrease in protein was the result of combined effects from reduced transcription and accelerated degradation of the existing protein pool in the cell. It is also not unusual that no significant correlation between mRNA and protein expression is found, since the protein levels in a tissue may depend on posttranslational modifications. GLI1 protein stability might be influenced by specific factors within the MuSCs microenvironment or during the transition from quiescence to the myogenic lineage. Post-translational modifications, such as phosphorylation, ubiquitination, or acetylation, could affect GLI1 protein stability. These modifications can regulate protein turnover and prevent its degradation. In the future, the specific signaling pathways and microenvironment cues that influence Gli1 mRNA and GLI1 protein expression in MuSCs during myogenic differentiation should be studied in more detail to elucidate the underlying regulatory mechanisms.

Lines 313: equal volume PBS were transplanted into INJURED TA muscle [...] instead of “equal volume PBS were transplanted into TA muscle [...]”?

Response: We thank the reviewer for the suggestion. We use “equal volume PBS were transplanted into injured TA muscle” instead of “equal volume PBS were transplanted into TA muscle” in revised line 313.